# Rethink Depth Separation with Intra-layer Links

## Abstract

The depth separation theory indicates that depth is significantly more powerful than width, which consists of two parts: i) there exists a function representable by a deep network; ii) such a function cannot be represented by a shallow network whose width is lower than a large threshold. However, the depth-width comparison therein is always based on the standard fully-connected networks, which motivates us to consider the question: Is width always significantly weaker than depth? Here, we report through bound estimation, explicit construction, and functional space analysis that adding shortcuts to connect neurons within a layer can greatly empower the width, such that a slender and shallow network can represent a deep network. Specifically, the width needed can be *exponentially* reduced by intra-layer links to represent the renowned "sawtooth" functions, compared to the threshold prescribed earlier. This means that width can also be powerful when armed with intra-layer links. Because the sawtooth function is a fundamental module in approximating polynomials and smooth functions, our saving of width is general for broader classes of functions. Lastly, the mechanism we identify can be translated into analyzing the expressivity of popular shortcut networks such as ResNet and DenseNet. We demonstrate that the addition of intra-layer links can also empower a ResNet to generate more linear pieces.

## 1 Introduction

Due to the widespread applications of deep networks in many important fields (LeCun et al., 2015), mathematically understanding the power of deep networks has been a central problem in deep learning theory (Poggio et al., 2020). The key issue is figuring out how expressive a deep network is or how increasing depth promotes the expressivity of a neural network better than increasing width. In this regard, there have been a plethora of studies on the expressivity of deep networks, which are collectively referred to as the *depth separation theory* (Safran et al., 2019; Vardi & Shamir, 2020; Gühring et al., 2020; Vardi et al., 2021; Safran & Lee, 2022; Venturi et al., 2022; Vardi et al., 2022).

A popular idea in depth separation theory is the complexity characterization that introduces appropriate complexity measures for functions represented by neural networks (Pascanu et al., 2013; Montufar et al., 2014; Telgarsky, 2015; Montúfar, 2017; Serra et al., 2018; Hu & Zhang, 2018; Xiong et al., 2020; Bianchini & Scarselli, 2014; Raghu et al., 2017; Sanford & Chatziafratis, 2022; Joshi et al., 2023), and then reports that increasing depth can greatly boost such a complexity measure. In contrast, a more tangible way to show the power of depth is to construct functions that can be expressed by a narrow network of a given depth, but cannot be approximated by shallower networks, unless its width is sufficiently large (Telgarsky, 2015; 2016; Arora et al., 2016; Eldan & Shamir, 2016; Safran & Shamir, 2017; Venturi et al., 2021). For example, (Eldan & Shamir, 2016) constructed a radial function that can be represented by a two-hidden-layer neural network with a poly number of neurons. But to achieve the same level of error, the exponential number of neurons is required for a one-hidden-layer neural network. (Telgarsky, 2015) employed a ReLU network to build a one-dimensional "sawtooth" function whose number of pieces scales exponentially over the depth. As such, a deep network can construct a sawtooth function with many pieces, while a shallow network cannot unless it is exponentially wide. (Arora et al., 2016) derived the upper bound of the maximal number of pieces for a univariate ReLU network, and used this bound to elaborate the separation between a deep and a shallow network. In a broad sense, we summarize the elements of establishing a depth separation theorem as the following: i) there exists a function representable

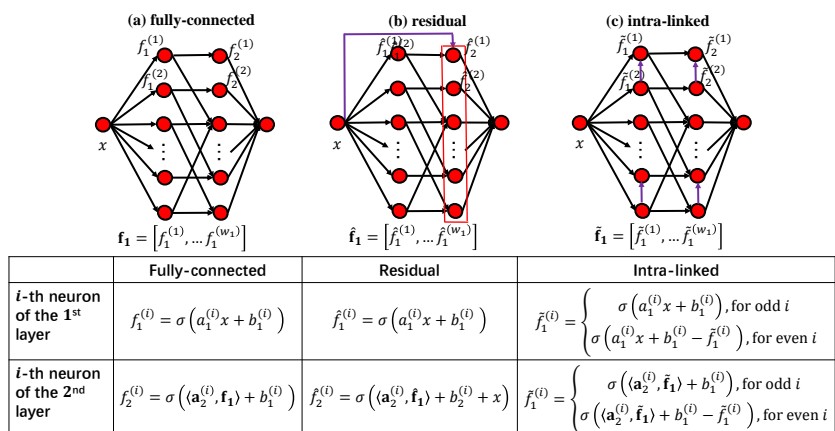

| | Fully-connected | Residual | Intra-linked |
|---|---|---|---|
| $i$-th neuron of the 1st layer | $f_1^{(i)} = \sigma\left(a_1^{(i)}x + b_1^{(i)}\right)$ | $\hat{f}_1^{(i)} = \sigma\left(a_1^{(i)}x + b_1^{(i)}\right)$ | $\tilde{f}_1^{(i)} = \begin{cases} \sigma\left(a_1^{(i)}x + b_1^{(i)}\right), \text{for odd } i \\ \sigma\left(a_1^{(i)}x + b_1^{(i)} - \tilde{f}_1^{(i)}\right), \text{for even } i \end{cases}$ |
| $i$-th neuron of the 2nd layer | $f_2^{(i)} = \sigma\left(\langle \mathbf{a}_2^{(i)}, \mathbf{f_1}\rangle + b_1^{(i)}\right)$ | $\hat{f}_2^{(i)} = \sigma\left(\langle \mathbf{a}_2^{(i)}, \hat{\mathbf{f}}_1\rangle + b_2^{(i)} + x\right)$ | $\tilde{f}_1^{(i)} = \begin{cases} \sigma\left(\langle \mathbf{a}_2^{(i)}, \tilde{\mathbf{f}}_1\rangle + b_1^{(i)}\right), \text{for odd } i \\ \sigma\left(\langle \mathbf{a}_2^{(i)}, \tilde{\mathbf{f}}_1\rangle + b_1^{(i)} - \tilde{f}_1^{(i)}\right), \text{for even } i \end{cases}$ |

Figure 1: (a) fully-connected, (b) residual, and (c) intra-linked (2-neuron linked). $x$ is a univariate input. In analogy to horizontal residual connections in ResNet, we take the intra-layer links as vertical residual connections. Inserting intra-layer links is essentially different from stacking layers in terms of the mechanism of generating new pieces, the number of (affine transforms, activation) being used, and the functional class.
by a deep network; ii) such a function cannot be represented by a shallow network whose width is lower than a large threshold.

It appears that depth is significantly more powerful than width, as indicated by the depth separation theory. However, their comparison is always based on the standard fully-connected neural networks. This motivates us to consider the following question: *Is width always significantly weaker than depth?* (Goyal et al., 2022) demonstrated that a wide network with only 12 layers can achieve performance on par with a deep network with 30 layers. In this paper, we also give a positive answer on width. We find that adding shortcuts to connect neurons within a layer can greatly empower the width, such that a slender and shallow network can represent a deep network. Our design is inspired by ResNet, which are shortcuts across layers (Figure 1(b)). The shallow network with residual connections can have comparable performance with a deep network, *i.e.*, residual connections can empower depth. Per structural symmetry, we embed shortcuts vertically, *i.e.*, linking neurons within a layer (Figure 1(c)) to force a neuron to take the outputs of other neurons in the same layer. From the perspective of the crossing number (Telgarsky, 2016), the non-symmetric structure of intra-layer linked networks can produce more oscillations than networks without intra-layer links. In this light, with intra-layer links, without the need to go exponentially wide, a shallow network can express as a complicated function as a deep network could, which means that the width can also be powerful, and its power can be stimulated by intra-layer links. Although intra-layer links are not popular in literature and practice, we find that using intra-layer links is attractive in terms of performance and parametric and computational complexity. Appendix H shows the encouraging preliminary regression and classification results of using intra-layer links solely and conjugated with residual links, respectively, on 5 synthetic datasets, 15 tabular datasets, and 2 image benchmarks.

Specifically, our roadmap to justify the power of width includes two milestones. 1) Through bound analysis and explicit construction, we substantiate that a network with intra-layer links can produce many more pieces than a fully-connected network, and the gain is at most exponential. This means that the intra-layer links can boost the expressive ability of a network. Furthermore, we empirically confirm the power of networks with intra-layer links with systematic experiments over a bunch of datasets. In addition, we highlight that the identified mechanism of generating more pieces in analyzing intra-layer links can be translated into other shortcut networks such as ResNet and DenseNet. We compare the number of pieces generated by the intra-linked network and ResNet. The upper bound of the intra-linked network is higher than ResNet, which suggests that adding intra-layer links can also boost the representation ability of ResNet. 2) Since intra-layer links are helpful in generating more pieces, they can empower a slender and shallow network to represent a function constructed by a deep network. We derive theorems (Theorems 10 and 12), showing that the threshold of width in depth separation theory can be lowered at a linear and exponential level. Remarkably, Theorem 12 is realized in representing the famous sawtooth function (Telgarsky, 2015). The sawtooth function, as a fundamental module in approximating polynomials and smooth functions (Yarotsky, 2017; Kidger & Lyons, 2020; Lu et al., 2021; Shen et al., 2021), is important in the approximation theory of deep learning. This implies that our result regarding saving the width can hold for a broader class of functions besides sawtooth functions.

To summarize, our contributions are threefold. 1) We point out the limitation of the current depth separation theory, which is always established on fully-connected networks. 2) We show via bound estimation and explicit construction that intra-layer links can make a ReLU network produce more pieces. 3) We provide a complimentary understanding of depth separation theory, demonstrating that width under an intra-link design can also be powerful.

## 2 RELATED WORK

A plethora of depth separation studies have shown the superiority of deep networks over shallow ones from perspectives of complexity analysis and constructive analysis.

The complexity analysis is to characterize the complexity of the function represented by a neural network, thereby demonstrating that increasing depth can greatly maximize such a complexity measure. Currently, one of the most popular complexity measures is the number of linear regions because it conforms to the functional structure of the widely-used ReLU networks. For example, (Pascanu et al., 2013; Montufar et al., 2014; Montúfar, 2017; Serra et al., 2018; Hu & Zhang, 2018; Hanin & Rolnick, 2019) estimated the bound of the number of linear regions generated by a fully-connected ReLU network by applying Zaslavsky's Theorem (Zaslavsky, 1997). (Xiong et al., 2020) offered the first upper and lower bounds of the number of linear regions for convolutional networks. Other complexity measures include classification capabilities (Malach & Shalev-Shwartz, 2019), Betti numbers (Bianchini & Scarselli, 2014), trajectory lengths (Raghu et al., 2017), global curvature (Poole et al., 2016), and topological entropy (Bu et al., 2020). Please note that using complexity measures to justify the power of depth demands a tight bound estimation. Otherwise, it is insufficient to say that shallow networks cannot be as powerful as deep networks, since deep networks cannot reach the upper bound.

The construction analysis is to find a family of functions that are hard to approximate by a shallow network, but can be efficiently approximated by a deep network. (Eldan & Shamir, 2016) built a special radial function that is expressible by a 3-layer neural network with a polynomial number of neurons, but a 2-layer network can do the same level approximation only with an exponential number of neurons. Later, (Safran & Shamir, 2017) extended this result to a ball function, which is a more natural separation result. (Venturi et al., 2021) generalized the construction of this type to a non-radial function. (Telgarsky, 2015; 2016) used an $\mathcal{O}(k^2)$-layer network to construct a sawtooth function. Given that such a function has an exponential number of pieces, it cannot be expressed by an $\mathcal{O}(k)$-layer network, unless the width is $\mathcal{O}(\exp(k))$. (Arora et al., 2016) estimated the maximal number of pieces a network can produce, and established the size-piece relation to advance the depth separation results from $(k^2, k)$ to $(k, k')$, where $k' < k$. (Daniely, 2017) proved that poly-size depth neural networks with (exponentially) bounded weights cannot approximate $f : \mathbb{S}^{d-1} \times \mathbb{S}^{d-1} \to \mathbb{R}$ which has the form $f(\mathbf{x}, \mathbf{x}') = g(\langle \mathbf{x}, \mathbf{x}' \rangle)$ whenever $g$ cannot be expressed by a low-degree polynomial. Other smart constructions include polynomials (Rolnick & Tegmark, 2017), functions of a compositional structure (Poggio et al., 2017), Gaussian mixture models (Jalali et al., 2019), and so on. Recently, (Malach & Shalev-Shwartz, 2019) explored the relationship between the expressive properties of a deep network and the trainability using gradient descent-based methods. (Vardi et al., 2022) proved that there are no functions that can be expressed by wide and shallow neural networks but cannot be approximated by a narrow but deep network. (Safran & Lee, 2022) extended the depth separation theory into the provable training guarantee by proving that a ball indicator cannot be learned by a shallow network but can be learned by a deeper network. (Ren et al., 2023) showed that a multi-layer neural network can be trained to learn the function $\mathrm{ReLU}(1 - \|\mathbf{x}\|)$ that cannot be approximated by any one-hidden-layer network. Our work also includes the construction, and we use an intra-linked network to efficiently build a sawtooth function.

## 3 NOTATION AND DEFINITION

**Notation 1** (Fully-connected networks and extra-linked networks). For an $\mathbb{R}^{w_0} \to \mathbb{R}$ ReLU DNN with widths $w_1, \ldots, w_k$ of $k$ hidden layers, we use $\mathbf{f}_0 = \left[ f_0^{(1)}, \ldots, f_0^{(w_0)} \right] = \mathbf{x} \in \mathbb{R}^{w_0}$ to denote the input of the network. Let $\mathbf{f}_i = \left[ f_i^{(1)}, \ldots, f_i^{(w_i)} \right] \in \mathbb{R}^{w_i}, i = 1, \cdots, k$, be the vector composed of outputs of all neurons in the $i$-th layer. The pre-activation of the $j$-th neuron in the $i$-th layer and the corresponding neuron is given by

$$g_i^{(j)} = \left\langle \mathbf{a}_i^{(j)}, \mathbf{f}_{i-1} \right\rangle + b_i^{(j)} \quad \text{and} \quad f_i^{(j)} = \sigma \left( g_i^{(j)} \right),$$

respectively, where $\sigma(\cdot)$ is the ReLU activation, and $\mathbf{a}_i^{(j)} \in \mathbb{R}^{w_{i-1}}, b_i^{(j)} \in \mathbb{R}$ are parameters. The output of this network is $g_{k+1} = \langle \mathbf{a}_k, \mathbf{f}_k \rangle + b_k$ for some $\mathbf{a}_k \in \mathbb{R}^{w_k}, b_k \in \mathbb{R}$. The extra-linked networks like ResNet and DenseNet are similar to the classical fully-connected networks except that the pre-activation of the current layer takes the outputs of some previous layers.

**Notation 2** (Intra-linked networks) For an $\mathbb{R}^{w_0} \to \mathbb{R}$ ReLU DNN with widths $w_1, \ldots, w_k$ of $k$ hidden layers, we now use the matrix $\mathbf{G}^i \in \mathbb{R}^{w_i \times w_i}$ to denote the connecting operations within the $i$-th hidden layer. If $\mathbf{G}_{p,q}^i \neq 0$, it means that the output of the $p$-th neuron is fed into the $q$-th neuron in the $i$-th hidden layer, and multiplied by a coefficient $\mathbf{G}_{p,q}^i$; otherwise, the output of the $p$-th neuron is not. $\mathbf{G}_{p,q \leq p}^i = 0$ by default, since there are no loops. Similar to the classical ReLU DNN, we use $\tilde{\mathbf{f}}_0 = \mathbf{x} \in \mathbb{R}^{w_0}$ and $\tilde{\mathbf{f}}_i = \left[ \tilde{\mathbf{f}}_i^{(1)}, \ldots, \tilde{\mathbf{f}}_i^{(w_i)} \right] \in \mathbb{R}^{w_i}$ to denote the input and the outputs of the $i$-th layer, respectively. The $j$-th pre-activation in the $i$-th layer and the output of the network are computed as the following:

$$g_i^{(j)} = \left\langle \mathbf{a}_i^{(j)}, \tilde{\mathbf{f}}_{i-1} \right\rangle + b_i^{(j)} \quad \text{and} \quad \tilde{f}_i^{(j)} = \sigma \left( g_i^{(j)} + \sum_{p < j} \mathbf{G}_{p,j}^i \tilde{f}_i^{(p)} \right)$$

for each $j$. Especially, we are interested in the case that every 2 neurons are linked in each layer ($\mathbf{G}_{2k-1,2k}^i \neq 0$, as Figure 1(c)) and the case that every neuron in a layer are linked by its preceding neurons ($\mathbf{G}^i$ is an upper-triangular matrix).

**Notation 3** (sawtooth functions and breakpoints) we say a piecewise linear (PWL) function $g : [a, b] \to \mathbb{R}$ is of "$N$-sawtooth" shape, if $g(x) = (-1)^{n-1} \left( x - (n-1) \cdot \frac{b-a}{N} \right)$, for $x \in \left[ (n-1) \cdot \frac{b-a}{N}, n \cdot \frac{b-a}{N} \right], n \in [N]$. We say $x_0 \in \mathbb{R}$ is a breakpoint of a PWL function $g$, if the left-hand and right-hand derivatives of $g$ at $x_0$ are not equal, *i.e.*, $g'_+(x) \neq g'_-(x)$.

Note that stacking layers is essentially different from inserting intra-layer links in terms of the basic mechanism of generating new pieces, the number of affine transforms, and the functional class. Therefore, adding intra-layer links is essentially different from increasing depth.

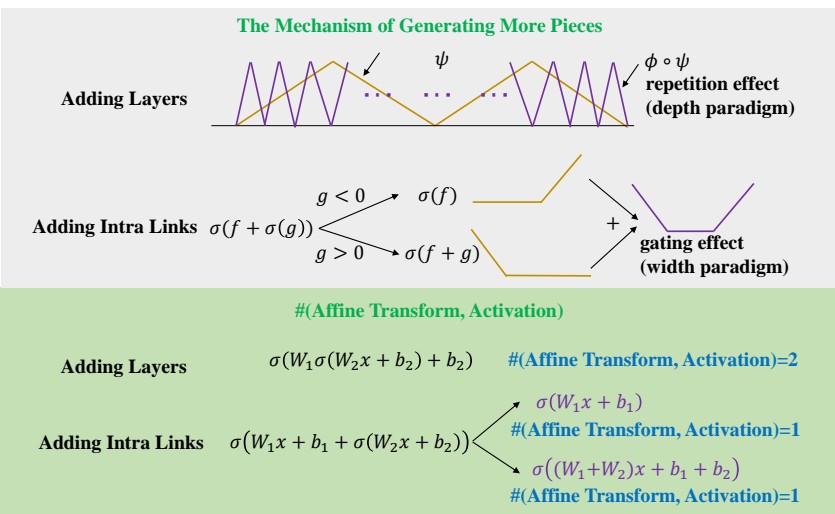

Figure 2: Adding intra-layer links is not equivalent to increasing depth in terms of the mechanism of generating more pieces, the number of (affine transforms, activation), and function classes.

● As Figure 2 shows, their mechanisms of producing pieces are fundamentally different. While the mechanism of adding a new layer is the repetition effect (multiplication), *i.e.*, when composing two layers that generate oscillation, each oscillation can generate more oscillations, which falls into the depth paradigm. The mechanism of intra-layer links is the gating effect (addition). The neuron being embedded has two activation states, and each state is leveraged to produce a breakpoint. Two states are combined to generate more pieces. Such a mechanism essentially conforms to the parallelism, which is of width paradigm. Therefore, adding intra-layer links does not increase depth.

● Adding intra-layer links does not increase the number of affine transforms and activation. As Figure 2 illustrates, a fully-connected network with two layers involves two times of affine trans-

formation and activation. In contrast, adding intra-layer links actually exerts a gating effect. When $\sigma(W_2 x + b_2) > 0$, the output is $\sigma((W_1 + W_2)x + b_1 + b_2)$; when $\sigma(W_2 x + b_2) = 0$, the output is $\sigma(W_1 x + b_1)$. The number of (affine transform, activation) is still one for both cases. We think the essence of depth is composition, which will lead to increased affine transforms and activations. Therefore, adding intra-layer links is different from stacking layers.

• The function classes represented by our intra-linked network and the deeper fully-connected network are not the same, either, and this will make a big difference. Given the same width, the deeper fully-connected network has a larger function class than a shallow intra-linked network. However, given the same width and depth, our intra-linked network has a larger function space (*i.e.*, number of pieces, VC dimension) than a fully-connected network (see results in Section 4.1). Therefore, essentially, an intra-linked network is a new type of network compared to fully-connected networks.

Defining the width and depth of a fully-connected network is straightforward. Because intra-linked networks are obtained by inserting intra-layer links into a standard fully-connected network, we define the width and depth of intra-linked networks to be the same as the width and depth of a fully-connected network resulting from removing intra-layer links layer by layer.

**Definition 1** (Width and depth of fully-connected networks (Arora et al., 2016))**.** *For any number of hidden layers $k \in \mathbb{N}$, input and output dimensions $w_0, w_{k+1} \in \mathbb{N}$, an $\mathbb{R}^{w_0} \to \mathbb{R}^{w_{k+1}}$ fully-connected network is given by specifying a sequence of $k$ natural numbers $w_1, w_2, \ldots, w_k$ representing widths of the hidden layers. The depth of the network is defined as $k + 1$, which is the number of (affine transforms, activation). The width of the network is $\max\{w_1, \ldots, w_k\}$.*

**Definition 2** (Width and depth of intra-linked networks (Fan et al., 2020))**.** *Given an intra-linked network $\Pi$, we delete the intra-layer links layer by layer to make the resultant network $\Pi'$ a standard fully-connected network, which means it has no isolated neurons and shortcuts. Then, we define the width and depth of $\Pi$ to be the same as the width and depth of $\Pi'$.*

## 4 RETHINK THE DEPTH SEPARATION WITH INTRA-LAYER LINKS

Our focus is the network using ReLU activation and the estimation of the number of pieces. The seminal depth separation theorems closest to us are the following:

**Theorem 1** (Depth separation $k^2$ vs $k$ (Telgarsky, 2015; 2016))**.** *For a natural number $k \geq 1$, there exists a sawtooth function representable by an $\mathbb{R} \to \mathbb{R}$ $(2k^2 + 1)$-layer fully-connected ReLU DNN of width $2$ such that if it is also representable by a $(k + 1)$-layer fully-connected ReLU DNN, this $(k + 1)$-layer fully-connected ReLU DNN should at least have the width of $2^k - 1$.*

**Theorem 2** (Depth separation $k$ vs $k'$ (Arora et al., 2016))**.** *For every pair of natural numbers $k \geq 1, w \geq 2$, there exists a function representable by an $\mathbb{R} \to \mathbb{R}$ $(k + 1)$-layer fully-connected ReLU DNN of width $w$ such that if it is also representable by a $(k' + 1)$-layer fully-connected ReLU DNN for any $k' \leq k$, this $(k' + 1)$-layer fully-connected ReLU DNN has width at least $\frac{1}{2} w^{\frac{k}{k'}}$.*

The above two theorems reveal that increasing depth can make a ReLU network express a much more complicated function than increasing width, which is at the heart of depth separation. Here, we show that width is not always significantly weaker than depth. Our primary argument is that if intra-layer links are inserted, there exist slender and shallow networks that previously could not express some hard functions constructed by deep networks now can do the job. Our investigation consists of two parts. First, we theoretically illustrate that adding intra-layer links can greatly increase the number of pieces via bound estimation and explicit construction. We also empirically confirm the regression and classification performance of networks with intra-layer links via 5 synthetic datasets, 15 tabular datasets, and 2 image benchmarks (Appendix H). Then, adding intra-layer links can help a network represent complicated functions such as sawtooth functions, without the need to go as wide as before. Favorably, the width needed can be reduced exponentially, which means the width is not that weak when intra-layer links are configured.

### 4.1 INTRA-LAYER LINKS CAN INCREASE THE NUMBER OF PIECES

#### 4.1.1 UPPER BOUND ESTIMATION

**Lemma 3.** *Let $g : \mathbb{R} \to \mathbb{R}$ be a PWL function with $w + 1$ pieces, then the breakpoints of $f := \sigma(g)$ consist of two parts: some old breakpoints of $g$ and at most $w + 1$ newly produced breakpoints. Furthermore, $f$ has $w + 1$ new breakpoints if and only if $g$ has $w + 1$ distinct zero points.*

*Proof.* A direct calculus. □

**Theorem 4** (Upper bound of fully-connected and extra-linked networks). *Let $f : \mathbb{R} \to \mathbb{R}$ be a PWL function represented by an $\mathbb{R} \to \mathbb{R}$ ReLU fully-connected or extra-linked neural networks, whose depth is $k + 1$ and widths are $w_1, \ldots, w_k$ of $k$ hidden layers, respectively. Then $f$ has at most $\prod_{i=1}^{k} (w_i + 1)$ pieces.*

**Remark 1.** This bound is actually the univariate case of the bound: $\prod_{i=1}^{k} \sum_{j=0}^{n} \binom{w_i}{j}$, derived in (Montúfar, 2017) for $n$-dimensional inputs. In Appendix A, we offer constructions to show that this bound is achievable in a depth-bounded but width-unbounded network (depth=3) (Proposition 7) and a width-bounded (width=3) but depth-unbounded network (Proposition 8) in one-dimensional space. Previously, many bounds (Pascanu et al., 2013; Montufar et al., 2014; Montúfar, 2017; Xiong et al., 2020) on linear regions were derived, however, it is unknown whether these bounds are vacuous or tight, particularly for networks with more than one hidden layer. Determining the tightness of a bound is essential in analyzing the approximation ability of a deep network. What makes Propositions 7 and 8 special is that they for the first time substantiate that (Montúfar, 2017)'s bound is tight over an arbitrary three-layer network and deeper networks with small widths, which fills the gap of bound estimation, although these results are for the one-dimensional case.

Theorem 4 also sharpens the bound in (Arora et al., 2016). Previously, they computed the number of pieces produced by a network of depth $k+1$ and widths $w_1, \ldots, w_k$ as $2^{k+1} \cdot (w_1 + 1) w_2 \cdots w_k$. The reason why their bound has an exponential term is that when considering how ReLU activation increases the number of pieces, they repetitively computed the old breakpoints generated in the previous layer. Our Lemma 3 implies that the ReLU activation in fact cannot double the number of pieces of a PWL function.

**Lemma 5** (A corollary of Lemma 3). *Let $g_1, g_2 : \mathbb{R} \to \mathbb{R}$ be two PWL functions with $w$ breakpoints in total. $f_1 := \sigma(g_1)$ and $f_2 := \sigma(g_2 - f_1)$. Then the breakpoints of $f_2$ include three parts: some breakpoints of $g_2$, some breakpoints of $f_1$, and at most $2w + 2$ newly produced breakpoints. Furthermore, $f_2$ has $2w + 2$ new breakpoints if and only if $g_2 - f_1$ has $2w + 2$ distinct zero points.*

Let us illustrate why the intra-linked architecture can produce more pieces. Given two PWL functions $g_1$ and $g_2$ which has a total of $w$ breakpoints, in the fully-connected architecture, $\sigma(g_1)$ and $\sigma(g_2)$ have totally at most $3w + 2$ breakpoints, which contains at most $w$ old breakpoints of $g_1, g_2$ and at most $2w + 2$ newly produced breakpoints. However, in the intra-linked architecture, $\sigma(g_2 - \sigma(g_1))$ can produce more breakpoints because $\sigma(g_1)$ has two states: activated or deactivated. Then, $\sigma(g_1)$ and $\sigma(g_2 - \sigma(g_1))$ consist of at most $w$ old breakpoints of $g_1, g_2$ and $(w + 1) + (2w + 2) = 3w + 3$ new breakpoints.

**Theorem 6** (Upper bound of $n_i$-neuron intra-linked networks). *Let $f : \mathbb{R} \to \mathbb{R}$ be a PWL function represented by a ReLU DNN with depth $k + 1$, widths $w_1, \ldots, w_k$, and every $n_i$ neurons linked in the $i$-th layer as Figure 1(c). Assuming that $n_i$ can divide $w_i$ without remainder, $f$ has at most $\prod_{i=1}^{k} \left( \frac{2^{n_i} - 1}{n_i} w_i + 1 \right)$ pieces.*

*Proof.* For conciseness, we only consider the case $n_i = 2$. The general case is nothing but repeating the same analysis in each layer several times. We prove by induction on $k$. For the base case $k = 1$, we assume for every odd $j$, the neurons $\tilde{f}_1^{(j)}$ and $\tilde{f}_2^{(j+1)}$ are linked. The number of breakpoints of $\tilde{f}_1^{(j)}$, $j = 1, \ldots, w_1$, is at most $2 + (-1)^j$. Hence, the first layer yields at most $\frac{3}{2} w_1 + 1$ pieces. For the induction step, we assume that for some $k \geq 1$, any $\mathbb{R} \to \mathbb{R}$ ReLU DNN with every two neurons linked in each hidden layer, depth $k + 1$ and widths $w_1, \ldots, w_k$ of $k$ hidden layers produces at most $\prod_{i=1}^{k} \left( \frac{3}{2} w_i + 1 \right)$ pieces. Now we consider any $\mathbb{R} \to \mathbb{R}$ ReLU DNN with every two neurons linked in each hidden layer, depth $k + 2$ and widths $w_1, \ldots, w_{k+1}$ of $k + 1$ hidden layers. By the induction hypothesis, each $\tilde{g}_{k+1}^{(j)}$ has at most $\prod_{i=1}^{k} \left( \frac{3}{2} w_i + 1 \right) - 1$ breakpoints. Then the breakpoints of $\sigma(\tilde{g}_{k+1}^{(j)})$ consist of some breakpoints of $\tilde{g}_{k+1}^{(j)}$ and at most $\prod_{i=1}^{k} \left( \frac{3}{2} w_i + 1 \right)$ newly generated breakpoints. Then $\tilde{g}_{k+1}^{(j+1)} - \tilde{f}_{k+1}^{(j)}$ has at most $2 \cdot \prod_{i=1}^{k} \left( \frac{3}{2} w_i + 1 \right) - 1$ breakpoints, based on Lemma 5. The breakpoints of $\tilde{f}_{k+1}^{(j+1)} = \sigma(\tilde{g}_{k+1}^{(j+1)} - \tilde{f}_{k+1}^{(j)})$ consist of some breakpoints of $\tilde{g}_{k+1}^{(j+1)} - \tilde{f}_{k+1}^{(j)}$ and at most $2 \cdot \prod_{i=1}^{k} \left( \frac{3}{2} w_i + 1 \right)$ newly generated breakpoints. Note that $\tilde{g}_{k+1}^{(1)}, \ldots, \tilde{g}_{k+1}^{(w_{k+1})}$ have totally at most $\prod_{i=1}^{k} \left( \frac{3}{2} w_i + 1 \right) - 1$ breakpoints. In all, the number of pieces we can therefore get is at most $1 + \frac{w_{k+1}}{2} \cdot \left( \prod_{i=1}^{k} \left( \frac{3}{2} w_i + 1 \right) + 2 \cdot \prod_{i=1}^{k} \left( \frac{3}{2} w_i + 1 \right) \right) + \prod_{i=1}^{k} \left( \frac{3}{2} w_i + 1 \right) - 1 = \prod_{i=1}^{k+1} \left( \frac{3}{2} w_i + 1 \right)$. □

**Remark 2**. Comparing Theorems 4 and 6, we surprisingly find that adding extra-layer links does not promote the upper bound, but adding intra-layer links does. This is not because this upper bound is vacuous (later, we show this bound is tight). Instead, adding previous layers into later layers does not produce new pieces in the maximal sense. In contrast, adding intra-layer links can improve the upper bound of fully-connected and extra-linked networks exponentially (dependent on depth), which means that intra-layer links can also empower ResNets. Thus, intra-layer links are a sufficiently interesting architecture in their sole existence or in synergy with extra-layer links.

**Theorem 7** (Upper Bound of Fully-connected Networks (Montúfar, 2017)). *Let $f : \mathbb{R}^n \to \mathbb{R}$ be a PWL function represented by an $\mathbb{R}^n \to \mathbb{R}$ ReLU DNN with depth $k + 1$ and widths $w_1, \ldots, w_k$ of $k$ hidden layers. Then $f$ has at most $\prod_{i=1}^{k} \sum_{j=0}^{n} \binom{w_i}{j}$ linear regions.*

**Theorem 8** (Upper Bound of Intra-linked Networks, proof in Appendix B). *Let $f : \mathbb{R}^n \to \mathbb{R}$ be a PWL function represented by an $\mathbb{R}^n \to \mathbb{R}$ ReLU DNN with every two neurons linked in each hidden layer, depth $k + 1$ and widths $w_1, \ldots, w_k$ of $k$ hidden layers. We assume each $w_i$ is even. Then $f$ has at most $\prod_{i=1}^{k} \sum_{j=0}^{n} \binom{\frac{3w_i}{2}+1}{j}$ linear regions.*

### 4.1.2 EXPLICIT CONSTRUCTION.

Despite that the bound estimation offers some hints, to convincingly illustrate that intra-layer links can increase the number of pieces, we need to supply the explicit construction for the intra-linked networks. The number of pieces in the construction should be bigger than the maximum a fully-connected network can achieve. Specifically, the constructions for 2-neuron intra-linked networks in Propositions 1 and 2 have a number of pieces larger than the upper bounds of fully-connected networks. In Proposition 3, by enumerating all possible cases, we present a construction for a 2-neuron intra-linked network of width 2 and arbitrary depth whose number of pieces is larger than what a fully-connected network of width 2 and arbitrary depth possibly achieves. Proposition 4 shows that $\prod_{i=1}^{k} \left( \frac{(w_i+1)w_i}{2} + 1 \right)$ pieces can be achieved by a one-hidden-layer all-intra-linked network. Propositions 5 and 6 provide rather tight constructions for an all-neuron intra-linked network of width 3&4 and arbitrary depth.

**Proposition 1** (The bound $\prod_{i=1}^{k} \left( \frac{3w_i}{2} + 1 \right)$ is tight for a two-hidden-layer 2-neuron intra-linked network, proof in Appendix C). *Given an $\mathbb{R} \to \mathbb{R}$ two-hidden-layer ReLU network, with every two neurons linked in each hidden layer, for any even $w_1 \geq 6, w_2 \geq 4$, there exists a PWL function represented by such a network, whose number of pieces is $\left( \frac{3w_1}{2} + 1 \right) \left( \frac{3w_2}{2} + 1 \right)$.*

**Proposition 2** (Use intra-linked networks to achieve $\prod_{i=1}^{k} \left( \frac{3w_i}{2} \right)$ pieces, proof in Appendix D). *There exists a $[0, 1] \to \mathbb{R}$ function represented by an intra-linked ReLU DNN with depth $k + 1$ and width $w_1, \ldots, w_k$ of $k$ hidden layers, whose number of pieces is at least $\frac{3w_1}{2} \cdot \ldots \cdot \frac{3w_k}{2}$.*

**Proposition 3** (Intra-layer links can greatly increase the number of pieces in an $\mathbb{R} \to \mathbb{R}$ ReLU network with width 2 and arbitrary depth, proof in Appendix E). *Let $f : \mathbb{R} \to \mathbb{R}$ be a PWL function represented by an $\mathbb{R} \to \mathbb{R}$ $(k + 1)$-layer ReLU DNN with widths 2 of all $k$ hidden layers. Then the number of pieces of $f$ is at most $\begin{cases} \sqrt{7}^k, & \text{if } k \text{ is even,} \\ 3 \cdot \sqrt{7}^{k-1}, & \text{if } k \text{ is odd.} \end{cases}$*

*There exists an $\mathbb{R} \to \mathbb{R}$ $(k + 1)$-layer 2-wide ReLU DNN, with neurons linked in each hidden layer, which can produce at least $7 \cdot 3^{k-2} + 2$ pieces.*

**Proposition 4** ( $\prod_{i=1}^{k} \left( \frac{(w_i+1)w_i}{2} + 1 \right)$ pieces for a one-hidden-layer all-neuron intra-linked network, proof in Appendix F). *Given an $\mathbb{R} \to \mathbb{R}$ one-hidden-layer ReLU network with all neurons linked in the hidden layer, there exists a PWL function represented by such a network, whose number of pieces is $\frac{(w_1+1)w_1}{2} + 1$.*

**Proposition 5** (An arbitrarily deep network of width=3 and with all neurons in each layer intra-linked $\mathbf{G}_{j-1,j}^i \neq 0$ can achieve at least $5^k$ pieces, proof in Appendix F). *There exists an $\mathbb{R} \to \mathbb{R}$ function represented by an intra-linked ReLU DNN with depth $k$, width 3 in each layer, and all neurons intra-linked in each layer, whose number of pieces is at least $5^k$.*

**Proposition 6** (An arbitrarily deep network of width=4 and with all neurons in each layer intra-linked $\mathbf{G}_{j-1,j}^i \neq 0$ can achieve at least $9^k$ pieces, proof in Appendix F). *There exists an $\mathbb{R} \to \mathbb{R}$ function represented by an intra-linked ReLU DNN with the depth $k$, width 4 in each layer, and all neurons in each layer intra-linked, whose number of pieces is at least $9^k$.*

**Remark 3.** Currently, intra-layer links are few studied in practice. (Zhang & Zhou, 2022) shows that using intra-layer links can improve the generalization of spiking networks. Concurrent to our work, (Sadat Shahir et al., 2023) confirms that using intra-layer links can result in a rapid convergence. Although the main focus of this draft is not to propose a new network architecture, we find that intra-layer links are a promising structure in terms of the balance between the performance and the parametric and computational complexity. To evaluate if intra-linked networks can deliver good performance as theory suggests, we compare intra-linked networks with fully-connected networks and ResNet on 5 synthetic datasets, 15 public tabular datasets, and 2 image benchmarks in Appendix H. Results show that on both regression and classification tasks, intra-linked networks can outperform the standard fully-connected network with fewer parameters, and using intra-layer links can also further enhance the performance of ResNet. At the same time, although intra-layer links have more links to optimize, our experiments also demonstrate that intra-linked networks do not suffer the optimization issue, which aligns with what was observed in (Sadat Shahir et al., 2023).

### 4.2 MODIFY THE DEPTH SEPARATION THEOREM WITH INTRA-LAYER LINKS

We summarize that the depth separation theorems consist of two elements: i) there exists a function representable by a deep network; ii) such a function cannot be represented by a shallow network whose width is lower than a threshold. Since adding intra-layer links can generally improve the capability of a network, if one adds intra-layer links to a shallow network, the function constructed by a deep network can be represented by a shallow network, even if the width of this shallow network is lower than the threshold. Theorem 10 showcases that a shallow network with all-neuron intra-layer links can save the width up to a linear reduction. Theorem 12 modify the depth separation ($k^2$ vs $k$) by presenting that a shallow network with intra-layer links only needs a linear width instead of an exponential width to express the sawtooth function. Since the exponential width is no longer needed, we contend that width is not significantly weaker than depth in this setting.

**Lemma 9** (A network with width=2 can approximate any univariate PWL function (Fan et al., 2021)). *Given an arbitrary univariate PWL function with $n$ pieces $p(x)$, there exists an $(n+1)$-layer network $\mathbf{D}(x)$ with two neurons in each layer such that $p(x) = \mathbf{D}(x)$.*

**Theorem 10** (Modify the depth separation $k^2$ vs 2). *For every $k \geq 2$, there exists a function $p(x)$ that can be represented by a $(k^2+1)$-layer ReLU DNN with 2 nodes in each layer, such that it cannot be represented by a classical 2-layer ReLU DNN $\mathbf{W}_2(x)$ with width less than $k^2-1$, but can be represented by a 2-layer, $(2k)$-wide intra-linked ReLU DNN $\tilde{\mathbf{W}}_2(x)$. The connecting operations of each layer are $\mathbf{G}_{j-1,j}^i = 1$, where $j < 2k$, and the rest entries are zeros.*

*Proof.* Combining Theorem 4, Proposition 4, and Lemma 9 straightly concludes the proof. □

**Lemma 11** (Representing sawtooth functions with intra-linked networks). *An intra-linked ReLU DNN of $k$ hidden layers with widths $w_1, \ldots, w_k$ can represent a sawtooth functions with $\prod_{i=1}^k 2^{w_i-2}$ pieces. The connecting operations within the $i$-th layer are that $\mathbf{G}_{<j,j}^i = 1$, $j \leq w_i - 2$, and the rest entries are zeros.*

*Proof.* It suffices to show the result for the first layer. Then the result follows from the composition of sawtooth functions. We set $f_1^{(1)} = \sigma(x - 1/2)$. Then the affine combination of $f_1^{(1)}$ and $y = x$, denoted as $h_1^{(1)}$, can be of sawtooth shape with $2^1$ pieces on [0,1], which gives $f_1^{(2)} = \sigma(h_1^{(1)} - 1/2)$. Similarly, the affine combination of $h_1^{(1)}$ and $f_1^{(2)}$, which is in fact the affine combination of $f_1^{(1)}, f_1^{(2)}$ and $y = x$, can be of sawtooth shape with $2^2$ pieces on $[0, 1]$. Following this procedure several times gives $f_1^{(3)}, \ldots, f_1^{(w_1-2)}$. Then the pre-activation of the second layer, as an affine combination of $f_1^{(1)}, \ldots, f_1^{(w_1-2)}$ and $y = x$ can be of sawtooth shape with $2^{w_1-2}$ pieces on $[0, 1]$. Remember that $y = x$ can be easily given by $f_1^{(w_1-1)} = \sigma(x)$ and $f_1^{(w_1)} = \sigma(-x)$. □

**Theorem 12** (Modify the depth separation $k^2$ vs $k$, the exponential saving of width). *For every $k \geq 1$, there is a $[0, 1] \to \mathbb{R}$ PWL function $p(x)$ represented by a fully-connected $(2k^2 + 1)$-layer ReLU DNN with at most $w$ nodes in each layer, such that it cannot be represented by a classical $(k+1)$-layer ReLU DNN $W_k(x)$ with width less than $w^k$, but can be represented by a $(k+1)$-layer intra-linked ReLU DNN $\tilde{W}_k(x)$ with width no more than $\log_2(w) \cdot k + 2$.*

*Proof.* Per (Telgarsky, 2016)'s construction, a fully-connected $(2k^2 + 1)$-layer ReLU DNN with at most $w$ nodes in each layer can produce a sawtooth function of $w^{k^2}$ pieces. Thus, it follows Theorem

4 that any classical $(k+1)$-layer ReLU DNN $W_k(x)$ with width less than $w^k - 1$ cannot generate $w^{k^2}$ pieces. However, according to Lemma 11, let $w_1 = w_2 = \cdots = w_k = \log_2(w) \cdot k + 2$, an intra-linked network can exactly express a sawtooth function with $w^{k^2}$ pieces. $\qquad\square$

From the perspective of the number of parameters, adding intra-layer links in a fully-connected network is no more than doubling the number of depth. When doubling the depth of a fully-connected network, the width is only reduced from $w$ to $w^{1/2}$. However, our saving for the width is exponential instead of polynomial, which means that adding intra-layer links has an essentially different mechanism from adding depth.

**Remark 4.** The depth-width comparison is a long-standing open problem. Indeed, the interplay between width and depth is intricate. In (Levine et al., 2020), it was shown that widening is necessary when deepening; otherwise, deepening becomes much more inefficient. The existing depth separation theory is primarily established for standard fully-connected networks. Here, we contend that adding intra-layer links mitigates the gap between width and depth in a linear and exponential manner. Note that our findings do not claim that width can replace depth. Instead, we derive a new relationship between width and depth in the context of shortcuts which is a more realistic setting, and can provide a different perspective than what is suggested in the depth separation theory. Because the sawtooth function is a fundamental module in deep learning approximation (Yarotsky, 2017; Kidger & Lyons, 2020; Lu et al., 2021; Shen et al., 2021), our saving of width is general for broader classes of functions such as smooth functions and polynomial functions.

## 5 DISCUSSION AND CONCLUSION

Well-established network architectures such as ResNet and DenseNet imply that incorporating shortcuts greatly empowers a neural network. However, only a limited number of theoretical studies attempted to explain the representation ability of shortcuts (Veit et al., 2016; Fan et al., 2021; Lin & Jegelka, 2018). Although intra-layer links and extra-layer links such as residual connections are essentially two different kinds of shortcuts, the techniques developed and the mechanisms identified in analyzing intra-linked networks can be extended to other networks with shortcuts. On the one hand, we identify conditions for the tightness of the bound, which has been proven to be stronger than existing results. Specifically, in the activation step, we distinguish the existing and newly generated breakpoints to avoid repeated counting, and then in the following pre-activation step, we maximize the oscillation to yield the most pieces after the next activation. On the other hand, the construction of functions in our work, *i.e.*, constructing oscillations by preserving existing breakpoints and splitting each piece into several ones, is generic in analyzing networks with extra-layer links, thereby explaining how they improve the representation power of a network. Earlier, we provide the bound for extra-linked networks such as ResNet and DenseNet, here we report our constructive results.

For example, it is straightforward to see that a one-neuron-wide ReLU DNN can represent PWL functions with at most three pieces, no matter how deep the network is. However, as Theorem 13 shows, with residual connections, a ResNet with $k$ neurons can represent a sawtooth function with $\mathcal{O}(k)$ pieces, which cannot be done by a fully-connected network. For DenseNet, Theorem 4 shows that an $\mathbb{R} \to \mathbb{R}$ ReLU DNN with depth $k+1$ and width $w_1, \ldots, w_k$ has at most $\prod_{i=1}^{k}(w_i + 1)$ pieces. If we add dense intra-layer links that connect any two neurons in a hidden layer to turn a fully-connected network into a DenseNet, Theorem 14 shows that the so-obtained DenseNet can produce many more pieces than the fully-connected network. The difference is exponential, *i.e.*, $1 + \prod_{i=1}^{k}\left(2^{w_i} - 1\right)$ vs $\prod_{i=1}^{k}(w_i + 1)$. The detailed proofs are put into Appendix G.

**Theorem 13.** *Let $f : \mathbb{R} \to \mathbb{R}$ be a PWL function represented by a one-neuron-wide ResNet. Mathematically, $f = c_{k+1}f_k + g_k$, where $g_1(x) = x, f_i = \sigma\left(a_i g_i + b_i\right), g_{i+1} = c_i f_i + g_i, c_{k+1}, a_i, b_i, c_i$ are parameters, for $i = 1, \ldots, k$. Then $f$ has at most $2^k$ pieces. Furthermore, this upper bound is tight and $f$ can be a sawtooth function with at most $2^k$ pieces.*

**Theorem 14.** *Let $f : \mathbb{R} \to \mathbb{R}$ be a PWL function represented by a DenseNet obtained by adding dense intra-layer links into a fully-connected network of $k$ hidden layers with widths $w_1, \ldots, w_k$. Then we can construct such a PWL function $f$ with at least $1 + \prod_{i=1}^{k}\left(2^{w_i} - 1\right)$ pieces.*

In this draft, via bound estimation (Theorems 6 and 8) and dedicated construction (Propositions 2, 3, 4, 5, and 6), we have shown that an intra-linked network is much more expressive than a fully-connected one, given the same width. Then, we have shown that a slender and shallow network that previously cannot express some functions constructed by deep networks now can do the job with intra-layer links (Theorems 10 and 12), suggesting that the width can also be powerful when aided

by intra-layer links. Meanwhile, the identified mechanism of generating pieces can also be used to decode the power of other shortcut networks such as ResNet and DenseNet. Future endeavors can be using intra-layer links to solve real-world problems.

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

## CONTENTS OF APPENDICES

Appendix A provides supplementary results to justify the tightness of Theorem 4.

Appendix B provides the proof of Theorem 8.

Appendix C provides the proof of Proposition 1.

Appendix D provides the proof of Proposition 2.

Appendix E provides the proof of Proposition 3.

Appendix F extends the results of 2-intra-linked networks to a network with more intra-layer links.

Appendix G demonstrates that the analysis tools developed for intra-linked networks can also be used to analyze the power of ResNet and DenseNet.

Appendix H demonstrates the empirical success of using intra-linked networks.

## A  SUPPLEMENTARY RESULTS FOR THE TIGHTNESS OF THEOREM 4

**Proposition 7** (The bound $\prod_{i=1}^{k}(w_i+1)$ is tight for a depth-bounded but width-unbounded network). *Given an $\mathbb{R} \to \mathbb{R}$ two-hidden-layer ReLU network, for any width $w_1 \geq 3, w_2 \geq 2$ in the first and second hidden layers, there exists a PWL function represented by such a network, whose number of pieces is $(w_1+1)(w_2+1)$.*

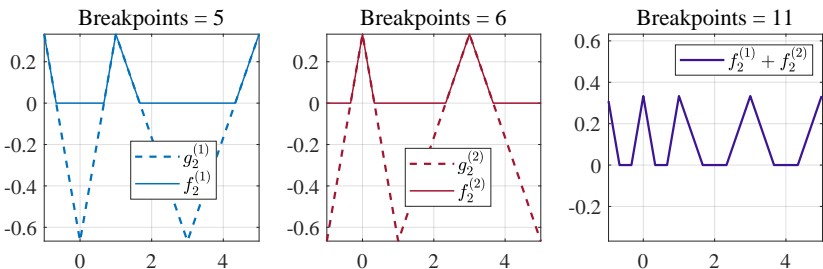

Figure 3: Construction of PWL functions to reach the bound of Proposition 7 when $w_1 = 3, w_2 = 2$.

*Proof.* To guarantee the bound $\prod_{i=1}^{k}(w_i+1)$ is tight, the following two requirements should be met: (i) each $g_i^{(j)}$, $i = 0, 1, 2$, $j = 1, \ldots, w_i$, has distinct zero points that are as many as its number of pieces, so that the activation step can produce the most new breakpoints; (ii) the breakpoints of each $g_{(i+1)}^{(j)}$, $i = 0, 1, 2$, $j = 1, \ldots, w_{i+1}$, as the affine combination of $\left\{f_i^{(1)}, \ldots, f_i^{(w_i)}\right\}$, contains all the breakpoints of $\left\{g_i^{(1)}, \ldots, g_i^{(w_i)}\right\}$, so that all the old breakpoints are reserved.

Now we give the proof in detail. Let $f_1^{(1)}(x) = \sigma(3x), f_1^{(2)}(x) = \sigma(-x+3), f_1^{(3)}(x) = \sigma\left(\frac{3}{2}x - \frac{3}{2}\right)$. When $w_1 = 3$, we set

$$g_2^{(1)} = -\left(f_1^{(1)} + f_1^{(2)} - f_1^{(3)} - 3 - \frac{1}{w_2+1}\right),$$
$$g_2^{(j)} = f_1^{(1)} + f_1^{(2)} - f_1^{(3)} - 3 - \frac{j}{w_2+1}, j = 2, \ldots, w_2.$$

When $w_1 > 3$, we let $f_1^{(j)} = \sigma(-2x - 2(j-3))$ and

$$g_2^{(1)} = -\left(f_1^{(1)} + f_1^{(2)} - f_1^{(3)} + \sum_{j=4}^{w_1}(-1)^{j-1}f_1^{(j)} - 3 - \frac{1}{w_2+1}\right)$$
$$g_2^{(j)} = f_1^{(1)} + f_1^{(2)} - f_1^{(3)} + \sum_{r=4}^{w_1}(-1)^{r-1}f_1^{(r)} - 3 - \frac{j}{w_2+1}, j = 2, \ldots, w_2.$$

Then $g_2^{(j)}$ has $w_1 + 1$ distinct zero points. Hence for $j = 1, \ldots, w_2$, the breakpoints of $f_2^{(j)} = \sigma\left(g_2^{(j)}\right)$ keeps all breakpoints of $g_2^{(j)}$ and yields $w_1 + 1$ new breakpoints. Note that $f_2^{(j)}$ and $f_2^{(j)}$ do not share new breakpoints, and $f_2^{(1)}$ and $f_2^{(2)}$ covers all the breakpoints of $\left\{g_2^{(j)}\right\}_{j=1}^{w_2}$. Therefore, the total number of pieces via an affine combination of $f_2^{(1)}, \ldots, f_2^{(w_2)}$ is $(w_1+1)(w_2+1)$ pieces.  $\square$

**Proposition 8** (The bound $\prod_{i=1}^{k}(w_i+1)$ is tight for a width-bounded but depth-unbounded network). *Given an $\mathbb{R} \to \mathbb{R}$ ReLU network with width $w$ for the first layer and 3 for other layers, for any depth $k \geq 2$, there exists a PWL function represented by such a network, whose number of pieces is $(w+1) \cdot 4^{k-1}$.*

*Proof.* Let $f_1^{(1)}, \ldots f_1^{(w)}$ be the same as in Proposition 7. Let

$$\tilde{g}_2 = \begin{cases} f_1^{(1)} + f_1^{(2)} - f_1^{(3)} - 3, & \text{if } w = 3, \\ f_1^{(1)} + f_1^{(2)} - f_1^{(3)} + \sum_{j=4}^{w}(-1)^{j-1}f_1^{(j)} - 3, & \text{if } w > 3. \end{cases}$$

We set

$$f_2^{(1)} = \sigma\left(2\tilde{g}_2 - \tfrac{1}{3}\right),$$
$$f_2^{(2)} = \sigma\left(-\tilde{g}_2 + \tfrac{2}{3}\right),$$
$$f_2^{(3)} = \sigma\left(\tfrac{3}{2}\tilde{g}_2 - \tfrac{1}{2}\right).$$

Now we continue our proof by induction. Assume that we have constructed $f_i^{(1)}$, $f_i^{(2)}$ and $f_i^{(3)}$, $i \geq 2$, then we set

$$\tilde{g}_{i+1} = f_i^{(1)} + f_i^{(2)} - f_i^{(3)} - \frac{3}{6^i}$$

and

$$f_{i+1}^{(1)} = \sigma\left(2\tilde{g}_{i+1} - \tfrac{2}{6^i}\right),$$
$$f_{i+1}^{(2)} = \sigma\left(-\tilde{g}_{i+1} - \tfrac{4}{6^i}\right),$$
$$f_{i+1}^{(3)} = \sigma\left(\tilde{g}_{i+1} + \tfrac{3}{6^i}\right).$$

Through a direct calculus, we know $\tilde{g}_{i+1}$ has $(w+1) \cdot 4^{i-1}$ pieces with opposite slopes in every two adjoint pieces and ranges from $0$ to $3/6^i$ in each piece except the leftmost and rightmost pieces, which implies we can obtain a total of $(w+1) \cdot 4^{k-1}$ pieces. $\qquad\square$

## B    PROOF OF THEOREM 8

**Lemma 15** (Zaslavsky's Theorem (Zaslavsky, 1975; Stanley, 2004)). *Let $\mathcal{A} = \{H_i \subset V : 1 \leq i \leq m\}$ be an arrangement in $\mathbb{R}^n$. Then, the number of regions for the arrangement $\mathcal{A}$ satisfies*

$$r(\mathcal{A}) \leq \sum_{i=0}^{n}\binom{m}{i}. \tag{1}$$

*Proof.* We prove by induction on $k$. For the base case $k = 1$, $\tilde{f}_1^{(2i-1)} = \sigma\left(\tilde{g}_1^{(2i-1)}\right)$ produces one hyperplane in the input space $\mathbb{R}^n$. Furthermore, $\tilde{f}_1^{(2i)} = \sigma\left(\tilde{g}_1^{(2i)} - \tilde{f}_1^{(2i-1)}\right) = \sigma\left(\tilde{g}_1^{(2i)} - \sigma\left(\tilde{g}_1^{(2i-1)}\right)\right)$ produces at most two hyperplanes in the input space $\mathbb{R}^n$. Therefore, in total, the $w_1$ neurons in the first layer produces $(1+2) \cdot \frac{w_1}{2} = \frac{3w_1}{2}$ hyperplanes in the input space $\mathbb{R}^n$. Then by Zaslavsky's Theorem, it will produce at most $\sum_{j=0}^{n}\binom{w_1+1}{j}$ linear regions in the input space $\mathbb{R}^n$. For the induction step, we assume that for some $k \geq 1$, any $\mathbb{R}^n \to \mathbb{R}$ ReLU DNN with every two neurons linked in each hidden layer, depth $k+1$ and widths $w_1, \ldots, w_k$ of $k$ hidden layers produces at most $\prod_{i=1}^{k}\sum_{j=0}^{n}\binom{\frac{3w_i}{2}+1}{j}$ linear regions. Now we consider any $\mathbb{R}^n \to \mathbb{R}$ ReLU DNN with every two neurons linked in each hidden layer, depth $k+2$ and widths $w_1, \ldots, w_{k+1}$ of $k+1$ hidden layers. Then for each linear region $S$ produced by the first $k+1$ layers, again, $\tilde{f}_{k+1}^{(2i-1)} = \sigma\left(\tilde{g}_{k+1}^{(2i-1)}\right)$ produces one hyperplane in $S$. Furthermore, $\tilde{f}_{k+1}^{(2i)} = \sigma\left(\tilde{g}_{k+1}^{(2i)} - \tilde{f}_{k+1}^{(2i-1)}\right) = \sigma\left(\tilde{g}_{k+1}^{(2i)} - \sigma\left(\tilde{g}_{k+1}^{(2i-1)}\right)\right)$ produces at most two hyperplanes in the $S$. Therefore, in total, the $w_{k+1}$ neurons in the $k+1$ layer produces $(1+2) \cdot \frac{w_{k+1}}{2} = \frac{3w_{k+1}}{2}$ hyperplanes in $S$. Then by Zaslavsky's Theorem, it will produce at most $\sum_{j=0}^{n}\binom{w_{k+1}+1}{j}$ linear regions in $S$. Thus $f$ has at most $\prod_{i=1}^{k}\sum_{j=0}^{n}\binom{\frac{3w_i}{2}+1}{j}$ linear regions. $\qquad\square$

## C    PROOF OF PROPOSITION 1

*Proof.* To guarantee the bound $\prod_{i=1}^{k}\left(\frac{3w_i}{2}+1\right)$ is tight, the following two conditions should be satisfied: (i) $\tilde{g}_i^{(j)}$ and $\tilde{g}_i^{(j+1)} - \tilde{f}_i^{(j)}$ have as many zero points as possible so that $\sigma(\tilde{g}_i^{(j)})$ and $\sigma(\tilde{g}_i^{(j+1)} - \tilde{f}_i^{(j)})$ can produce the maximal number of breakpoints; (ii) all old breakpoints of $\left\{\tilde{g}_i^{(1)}, \ldots, \tilde{g}_i^{(w_i)}\right\}$ are reserved by $\tilde{g}_{i+1}^{(j)}$, an affine transform of $\left\{\tilde{f}_i^{(1)}, \ldots, \tilde{f}_i^{(w_i)}\right\}$.

We first consider the first hidden layer. Let

$$\tilde{f}_1^{(1)}(x) = \sigma\left(\tfrac{9}{2}x - 27\right), \quad \tilde{f}_1^{(2)}(x) = \sigma\left(\tfrac{3}{2}x - \tilde{f}_1^{(1)}(x)\right)$$
$$\tilde{f}_1^{(3)}(x) = \sigma(-2x + 2), \quad \tilde{f}_1^{(4)}(x) = \sigma\left(-x + 2 - \tilde{f}_1^{(3)}(x)\right)$$
$$\tilde{f}_1^{(5)}(x) = \sigma\left(-\tfrac{7}{2}x - \tfrac{7}{4}\right), \quad \tilde{f}_1^{(6)}(x) = \sigma\left(-2x + 8 - \tilde{f}_1^{(5)}(x)\right).$$

When $w_1 = 6$, we set $\tilde{g} = -\tfrac{2}{9}\tilde{f}_1^{(1)} - \tilde{f}_1^{(2)} + \tfrac{1}{2}\tilde{f}_1^{(3)} + \tilde{f}_1^{(4)} - \tfrac{4}{7}\tilde{f}_1^{(5)} - \tilde{f}_1^{(6)}$.

When $w_1 > 6$, for each odd $j > 6$, let $\tilde{f}_1^{(j)} = \sigma\left(-5\left(x - a_j + 3\right)\right), \tilde{f}_1^{(j+1)} = \sigma\left(-2\left(x - a_j\right) - \tilde{f}_1^{(j)}\right)$, where $a_j = -\tfrac{19}{2} - 9\left(\tfrac{j-1}{2} - 3\right)$, then the output of the first layer is expressed as the following: $\tilde{g} = -\tfrac{2}{9}\tilde{f}_1^{(1)} - \tilde{f}_1^{(2)} + \tfrac{1}{2}\tilde{f}_1^{(3)} + \tilde{f}_1^{(4)} - \tfrac{4}{7}\tilde{f}_1^{(5)} - \tilde{f}_1^{(6)} + \sum_{j=7, j \text{ is odd}}^{w_2}(-1)^{\frac{j+1}{2}}\left(\tfrac{2}{5}f_1^{(j)} + f_1^{(j+1)}\right)$, which has $\tfrac{3}{2}w_1 + 1$ pieces and whose adjacent pieces have slopes of opposite signs. Note that any line $y = b$, where $b \in (-13/2, -6)$, can cross all pieces of $\tilde{g} + b$. Thus, $g$ fulfills the conditions of Lemma 5. We divide the breakpoints of $\tilde{g}$ into two parts: $B_{upper} = \{x : x \text{ is a breakpoint of } \tilde{g} \text{ and } \tilde{g}(x) > b\}$ and $B_{lower} = \{x : x \text{ is a breakpoint of } \tilde{g} \text{ and } \tilde{g}(x) \le b\}$. We refer to their counts as $\#B_{upper}$ and $\#B_{lower}$.

$$\tilde{f}_2^{(1)} \qquad\qquad \tilde{f}_2^{(2)} \qquad\qquad \tilde{f}_2^{(3)} \qquad\qquad \tilde{f}_2^{(4)}$$

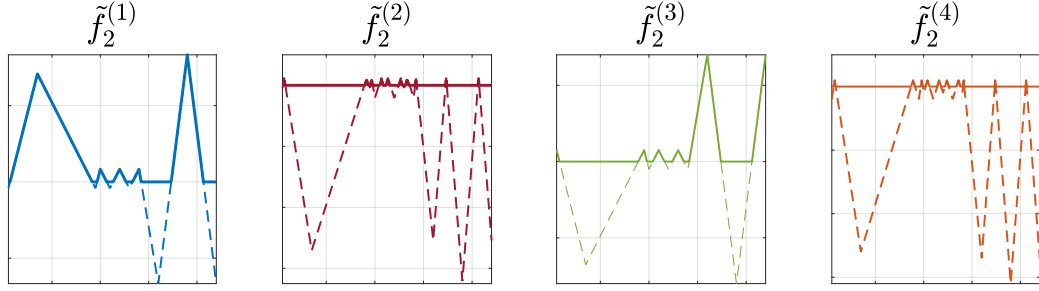

Figure 4: The PWL functions that reach the bound of Proposition 1 when $w_1 = 6$, $w_2 = 4$.

Next, we construct the second hidden layer. $\tilde{f}_2^{(1)} := \sigma\left(\tilde{g} + b_1\right)$, where $b_1 \in (-13/2, -6)$, has $\tfrac{3}{2}w_1 + 1$ new breakpoints. Then by choosing some scaling parameter $a \in (0, 1)$ bias $b_2$ to fulfill Lemma 5, we can also make $a\tilde{g} + b_2 - \tilde{f}_2^{(1)}$ has $3w_1 + 2$ distinct zero-points, which implies $\tilde{f}_2^{(2)} := \sigma\left(a\tilde{g} + b_2 - \tilde{f}_2^{(1)}\right)$ has $3w_1 + 2$ newly produced breakpoints. Therefore, the affine combination of $\tilde{f}_2^{(1)}$ and $\tilde{f}_2^{(2)}$ contains all breakpoints of $B_{upper}$, and has $\#B_{upper} + \left(\tfrac{3}{2}w_1 + 1\right) + (3w_1 + 2)$ breakpoints. To reserve all the breakpoints of $\tilde{g}$, we do the similar thing for $-\tilde{g}$ to gain $\tilde{f}_2^{(3)}$ and $\tilde{f}_2^{(4)}$, whose affine combination has $\#B_{lower} + \left(\tfrac{3}{2}w_1 + 1\right) + (3w_1 + 2)$ breakpoints, which contains all breakpoints in $B_{lower}$, and shares no breakpoints with the affine combination of $\left\{\tilde{f}_2^{(1)}, \tilde{f}_2^{(2)}\right\}$.

Hence, the affine combination of $\left\{\tilde{f}_2^{(1)}, \tilde{f}_2^{(2)}, \tilde{f}_2^{(3)}, \tilde{f}_2^{(4)}\right\}$ has $\#B_{upper} + \#B_{lower} + 2 \cdot \left(\tfrac{3w_1}{2} + 1\right) + 2 \cdot (3w_1 + 2) = \left(\tfrac{3w_1}{2}\right) + 6 \cdot \left(\tfrac{3w_1}{2} + 1\right)$ breaking points, which contains all the breakpoints of $\tilde{g}$. $\left\{\tilde{f}_2^{(1)}, \tilde{f}_2^{(2)}, \tilde{f}_2^{(3)}, \tilde{f}_2^{(4)}\right\}$ are visualized in Figure 4. Repeating this procedure by selecting different $b_1, a, b_2$, we can construct the remaining $\{\tilde{f}_2^{(i)}\}_{i=5}^{w_2}$ such that the affine transformation of $\{\tilde{f}_2^{(i)}\}_{i=1}^{w_2}$ has pieces of $\tfrac{3}{2}w_1 + \tfrac{3w_2}{2} \cdot \left(\tfrac{3}{2}w_1 + 1\right) + 1 = \left(\tfrac{3w_1}{2} + 1\right)\left(\tfrac{3w_2}{2} + 1\right)$. $\square$

## D   PROOF OF PROPOSITION 2

*Proof.* Let $\phi(x) = x$ defined over $[0, \Delta]$. The core of the proof is to use a one-hidden-layer network of $w \ge 2$ neurons to create $\tfrac{3w}{2}$ pieces from $\phi(x)$.

Let $\delta = \tfrac{2\Delta}{3w}$. Set $\tilde{g}^{(1)} = 3\phi - 3\delta$, $\tilde{f}^{(1)} = \sigma\left(\tilde{g}^{(1)}\right)$, $\tilde{g}^{(2)} = \phi$, $\tilde{f}^{(2)} = \sigma\left(\tilde{g}^{(2)} - \tilde{f}^{(1)} + \delta\right)$, and $\tilde{g}^{(2j+1)} = 4\phi - 4(3j + 1)\delta$, $f^{(2j+1)} = \sigma\left(\tilde{g}^{(2j+1)}\right)$, $\tilde{g}^{(2j+2)} = 2\phi - 6j\delta$, $\tilde{f}^{(2j+2)} = $

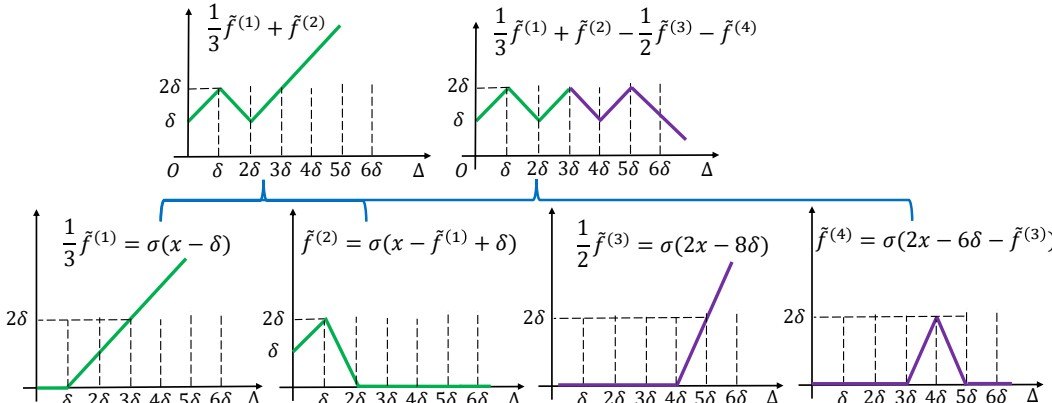

Figure 5: A schematic illustration of how to use an intra-linked network to generate a sawtooth function.

$\sigma\left(\tilde{g}^{(2j+2)} - \tilde{f}^{(2j+1)}\right)$, for all $j = 1, \ldots, w/2 - 1$. The output of this one-hidden-layer network is $\xi_{\Delta,w}(x) = \frac{1}{3}\tilde{f}^{(1)} + \tilde{f}^{(2)} - \delta + \sum_{j=1}^{\frac{w}{2}-1}(-1)^j\left(\frac{1}{2}\tilde{f}^{(2j+1)} + \tilde{f}^{(2j+2)}\right)$, which has $\frac{3w}{2}$ pieces on $[0, \Delta]$. $\xi_{\Delta,w}(x)$ is of slope $(-1)^j$ on $[j\delta, (j+1)\delta]$, $j = 0, \ldots, 3w/2 - 1$, and ranges from 0 to $\delta$ on each piece. Figure 5 shows how the affine transformation of $\{\tilde{f}^{(1)}, \tilde{f}^{(2)}, \tilde{f}^{(3)}, \tilde{f}^{(4)}\}$ constructs a sawtooth function of 6 pieces. Please note that flipping $\phi(x)$ or translating $\phi(x)$ will not prevent $\xi_{\Delta,w}(\phi(x))$ from generating $\frac{3w}{2}$ pieces.

The targeted intra-linked ReLU network with depth $k + 1$ and width $w_1, \ldots, w_k$ of $k$ hidden layers is designed as $\xi_{\Delta_k,w_k} \circ \xi_{\Delta_{k-1},w_{k-1}} \circ \cdots \circ \xi_{\Delta_1,w_1}(x)$, where $\Delta_i = 1/\left(\prod_{j=1}^{i-1} \frac{3w_i}{2}\right)$. □

## E    PROOF OF PROPOSITION 3

*Proof.* For the first assertion, we claim that each pre-activation $g_i^{(j)}$, $2 \leq i \leq k$, $j = 1, 2$, cannot make its two adjacent pieces have slopes with different signs, which implies the pre-activation cannot produce the most breakpoints as in Lemma 3. In fact, $g_2^{(j)}$, $j = 1, 2$, has at most 3 pieces. If some $g_2^{(j)}$ has 3 pieces, then by enumeration, we know either it has a 0 slope, or it has two adjacent pieces with slopes of the same sign (see Figure 6). Hence, $f_2^{(j)}$, $j = 1, 2$, has at most 2 new breakpoints. Then the output of the second layer has at most $2 + 2 \times 2 = 6$ breakpoints and 7 pieces. Applying a similar method to each piece, we can finish the proof via a simple induction step.

Now we come to the second assertion. For convenience, we say an $\mathbb{R} \to \mathbb{R}$ PWL function $f$ is of "triangle-trapezoid-triangle" shape on $[a, b] \subset \mathbb{R}$, if there exists a partition of $[a, b]: a < x_1 < x_2 < \cdots < x_6 < b$ and a positive constant $c$, such that

$$f(x) = \begin{cases} c, & \text{if } x = a, x_2, x_5, b \\ -c, & \text{if } x = x_1, x_6 \\ -3c, & \text{if } x \in [x_3, x_4] \\ \text{linear connection}, & \text{otherwise.} \end{cases}$$

Given a PWL function $f : \mathbb{R} \to \mathbb{R}$ of "triangle-trapezoid-triangle" shape on $[a, b]$, with a partition $a < x_1 < x_2 < \cdots < x_6 < b$ and $f(a) = c > 0$, if we set

$$\begin{aligned} g^{(1)} &= 4f, \\ g^{(2)} &= 2f - \frac{3c}{2}, \\ f^{(1)} &= \sigma\left(g^{(1)}\right), \\ f^{(2)} &= \sigma\left(g^{(2)} - f^{(1)}\right), \end{aligned}$$

then $g = -\frac{1}{4}f^{(1)} + f^{(2)} + \frac{c}{8}$ is of "triangle-trapezoid-triangle" shape on $[a, x_2]$, $[x_2, x_5]$, and $[x_5, b]$, respectively.

Using this fact, we can construct a PWL function represented by a $(k + 1)$-layer 2-neuron wide intra-linked ReLU DNN, which has $7 \cdot 3^{k-2} + 2$ pieces. Actually, if we set

$$\tilde{f}_1^{(1)} = \sigma(2x),$$
$$\tilde{f}_1^{(2)} = \sigma\left(x - \tilde{f}_1^{(1)} + 1\right),$$
$$\tilde{g}_2^{(1)} = -4\tilde{f}_1^{(2)} + 2,$$
$$\tilde{g}_2^{(2)} = -2\tilde{f}_1^{(2)} + \frac{3}{2},$$

then through a direct calculus, $\frac{1}{4}\tilde{f}_2^{(1)} + \tilde{f}_2^{(2)} - \frac{3}{8}$ is of "triangle-trapezoid-triangle" shape on $[-1, 1]$. Using the fact above repeatedly, we construct a PWL function represented by an $\mathbb{R} \to \mathbb{R}$ $(k+1)$-layer, 2-wide, intra-linked ReLU DNN, which is constant on $(-\infty, -1] \cup [1, \infty)$ and of "triangle-trapezoid-triangle" shape on $\left[-1 + \frac{2n}{3^{k-2}}, -1 + \frac{2(n+1)}{3^{k-2}}\right]$, $n = 0, \ldots, 3^{k-2} - 1$.

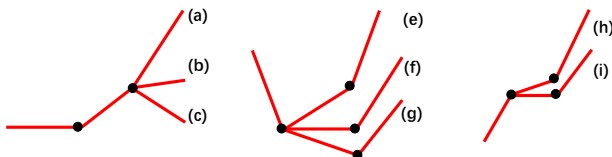

Figure 6: Enumerating all possible shapes of $g_2^{(j)}$, $j = 1, 2$ in Proposition 3.

$\square$

# F    EXTENSION TO MORE INTRA-LAYER LINKS

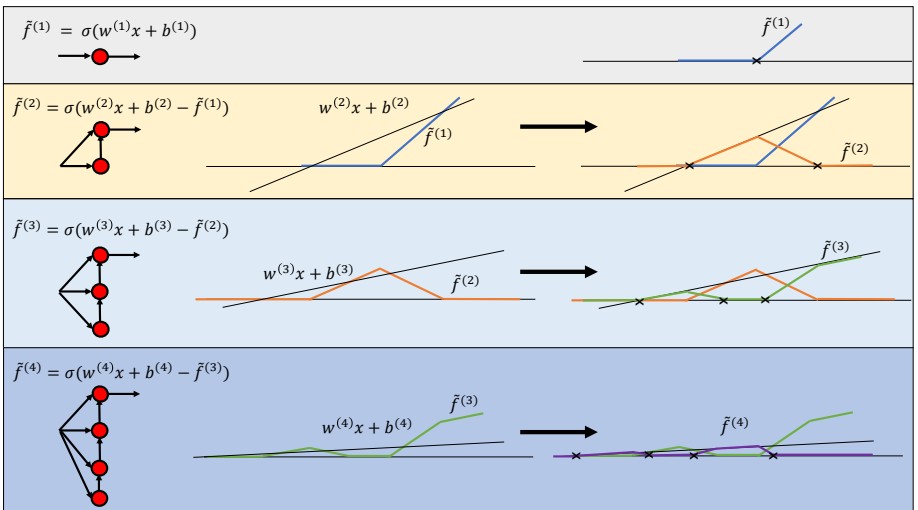

Figure 7: The construction demonstrating that the bound $\prod_{i=1}^{k}\left(\frac{(w_i+1)w_i}{2} + 1\right)$ is tight for a one-hidden-layer intra-linked network.

## F.1    PROOF OF PROPOSITION 4

*Proof.* Without loss of generality, a one-hidden-layer network with all neurons intra-linked is mathematically formulated as the following:

$$\begin{cases} \tilde{f}^{(1)} = \sigma(w^{(1)}x + b^{(1)}) \\ \tilde{f}^{(j+1)} = \sigma(w^{(j)}x + b^{(j)} - \tilde{f}^{(j)}) \end{cases}. \tag{2}$$

To prove that the bound $\prod_{i=1}^{k}\left(\frac{(w_i+1)w_i}{2} + 1\right)$ is tight for a one-hidden-layer network, the key is to make each $\tilde{f}^{(j)}$ produce $j$ new breakpoints and have $j$ non-zero pieces that share a point with $y = 0$.

We use mathematical induction to derive our construction. Figure 7 schematically illustrates the key idea in our construction.

First, let $\tilde{f}^{(1)} = \sigma(x+1)$ and $\tilde{f}^{(2)} = \sigma(0.5 \times (x+2) - \tilde{f}^{(1)})$. Note that $\tilde{f}^{(1)}$ has 1 non-zero piece that shares a point with $y = 0$, and $\tilde{f}^{(2)}$ has 2 non-zero pieces that share a common point with $y = 0$.

Then, given $\tilde{f}^{(j)}, j \geq 2$, we suppose $\tilde{f}^{(j)}$ has $j$ non-zero pieces that share a point with $y = 0$. Since $\tilde{f}^{(j)}$ is continuous, we select its peaks $\{(x_{p_i}, \tilde{f}^{(j)}(x_{p_i}))\}$ by the following conditions: i) $\tilde{f}^{(j)}$ is not differentiable at $x_{p_i}$; ii) $\tilde{f}^{(j)}(x_{p_i}) \neq 0$. Next, let $(x^*, \tilde{f}^{(j)}(x^*))$ be the lowest peak of $\tilde{f}^{(j)}$. As long as the slope $w^{(j+1)}$ and the bias $b^{(j+1)}$ satisfy

$$\begin{cases} w^{(j+1)} < \frac{\tilde{f}_1^{(j)}(x^*)}{x^*+j+1} \\ b^{(j+1)} = w_{j+1} \times (j+1) \end{cases}, \tag{3}$$

$w^{(j+1)}x + b^{(j+1)}$ crosses and only crosses $j$ pieces of $\tilde{f}^{(j)}$. These pieces are exactly non-zero pieces that share a point with $y = 0$. Thus, plus the breakpoint $-\frac{b^{(j+1)}}{w^{(j+1)}}$, $\tilde{f}^{(j+1)}$ generates a total of $j+1$ new breakpoints. At the same time, $\tilde{f}^{(j+1)}$ has $j+1$ non-zero pieces that share a point with $y = 0$. Figure 7 illustrates the process of induction.

Finally, the total number of breakpoints is $\sum_{j=1}^{w_1} j = \frac{(w_1+1)w_1}{2}$, which concludes our proof.

$\square$

### F.2 PROOF OF PROPOSITION 6

*Proof.* The core of the proof is to use a one-hidden-layer all-neuron-intra-linked network of width 4 to create a quasi-sawtooth function with as many pieces as possible. We construct four neurons as follows:

$$\begin{cases} \tilde{f}^{(1)} = \sigma(2x) \\ \tilde{f}^{(2)} = \sigma(x + 1 - \sigma(\tilde{f}^{(1)})) \\ \tilde{f}^{(3)} = \sigma(\frac{1}{3}(x+2) - \tilde{f}^{(2)}) \\ \tilde{f}^{(4)} = \sigma(\frac{1}{9}(x+3) - \tilde{f}^{(3)}) \end{cases}. \tag{4}$$

The profiles of $\tilde{f}^{(1)}, \tilde{f}^{(2)}, \tilde{f}^{(3)}, \tilde{f}^{(4)}$ are shown in Figure 8(a).

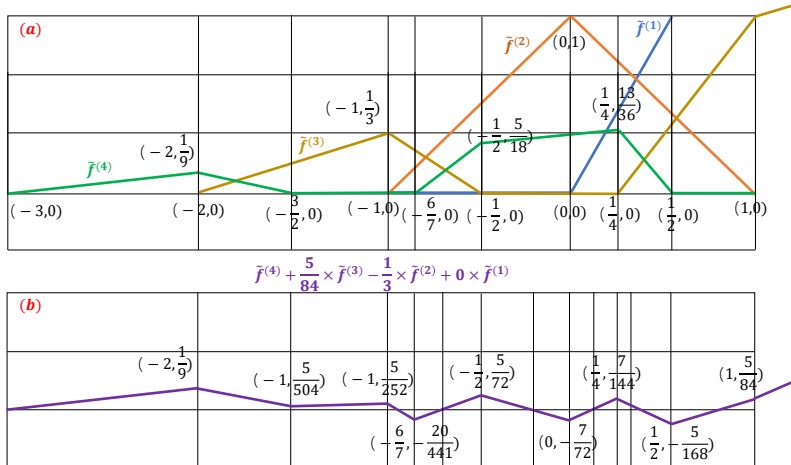

Figure 8: A schematic illustration of how to use an intra-linked network to generate a sawtooth function.

By combining $\tilde{f}^{(1)}, \tilde{f}^{(2)}, \tilde{f}^{(3)}, \tilde{f}^{(4)}$ with carefully calibrated coefficients, we have the following quasi-sawtooth function that has 9 pieces are

$$\eta(x) = \tilde{f}^{(4)} + \frac{5}{84} \times \tilde{f}^{(3)} - \frac{1}{3} \times \tilde{f}^{(2)} + 0 \times \tilde{f}^{(1)}. \tag{5}$$

As shown in Figure 8(b), we have marked all breakpoints of $\eta(x)$ to validate its correctness.

Next, we just need to let each layer of the intra-linked network represent a stretched and down-pulled variant of $\eta(x)$, *e.g.*, the $k$-th layer $\eta_k(x) = M_k \cdot \eta(x) - B_k$, where $M_k$ is a sufficiently large number and $B_k > \frac{5}{504} M_k + 3$ to ensure that $[-3, 0]$ is within the function range of $\eta_k(x)$.

Finally, the constructed network is

$$\eta_k \circ \eta_{k-1} \circ \cdots \circ \eta_1(x). \tag{6}$$

$\square$

### F.3 Proof of Proposition 5

*Proof.* Following the same spirit in proof of Proposition 6, we construct three neurons as follows:

$$\begin{cases} \tilde{f}^{(1)} = \sigma(2x) \\ \tilde{f}^{(2)} = \sigma(x + 1 - \sigma(\tilde{f}^{(1)})) \\ \tilde{f}^{(3)} = \sigma(\frac{1}{3}(x + 2) - \tilde{f}^{(2)}) \end{cases}. \tag{7}$$

The target function that returns us 5 pieces is

$$\xi(x) = \frac{1}{100} \times \tilde{f}^{(3)} - \frac{1}{3} \times \tilde{f}^{(2)} + 0 \times \tilde{f}^{(1)}. \tag{8}$$

Next, we just need to let each layer of the intra-linked network represent a stretched and down-pulled variant of $\xi(x)$, *e.g.*, the $k$-th layer $\xi_k(x) = T_k \cdot \eta_k(x) - C_k$, where $T_k$ is a sufficiently large number and $C_k > \frac{1}{200} T_k + 2$ to ensure that $[-2, 0]$ is within the function range of $\xi_k(x)$.

Finally, the constructed network is

$$\xi_k \circ \xi_{k-1} \circ \cdots \circ \xi_1(x). \tag{9}$$

$\square$

## G  Analysis Extended to ResNet and DenseNet

### G.1  Proof of Theorem 13

*Proof.* We set $c_i = -2$ and $a_i = 1$ for all $i$ and set $b_1 = 0$, $b_i = 2 - 2^{-i+2}$ for $i = 2, \ldots, k$. $\square$

### G.2  Proof of Theorem 14

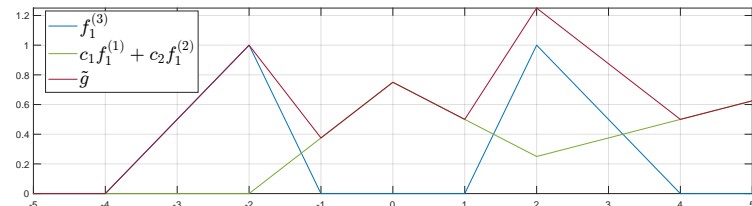

Figure 9: A schematic illustration of using constructions in intra-linked networks to analyze DenseNet.

*Proof.* For convenience, we only consider the case $w_i = 3$, the general case is just simply repeating this procedure. Let

$$f_1^{(1)}(x) = \sigma(3x), f_1^{(2)}(x) = \sigma\left(\frac{3}{2}x + 3 - f_1^{(1)}\right), f_1^{(3)}(x) = \sigma\left(\frac{x}{2} + 2 - f_2 - \frac{1}{3}f_1\right)$$

be the neurons of the first hidden layer. Then we consider the affine combination. It is easy to find coefficients $c_1, c_2$ such that

$$c_1 f_1^{(1)}(x) + c_2 f_1^{(2)}(x) = \begin{cases} \frac{3}{8}x + \frac{3}{4}, & x \in [-2, 0], \\ -\frac{1}{4}x + \frac{3}{4}, & x \in [0, 2], \\ \frac{x}{8}, & x \in [2, 4], \\ 0, & \text{otherwise.} \end{cases}$$

A direct calculus gives $\tilde{g} = c_1 f_1^{(1)}(x) + c_2 f_1^{(2)}(x) + f_1^{(3)}(x)$, shown in Figure 9. Then following the same idea in analyzing intra-linked networks, we know all the non-constant pieces of $\tilde{g}$ can generate $\prod_{i=1}^{k} (2^{w_2} - 1)$ in the next layer and the result follows after a simple induction step. $\qquad\square$

Theorems 13 and 14 confirm that adding simple links can greatly improve the representation ability of a network. Actually, both ResNet and intra-layer linked networks do not increase the number of parameters a lot, but they can represent more complicated functions than the fully-connected of the same width in each layer. Hence, the linked structure can improve the efficiency of parameters. Besides, we can see from proofs of Theorems 13 and 14 that the idea and construction in analyzing intra-linked networks can indeed be utilized to analyze other important architectures.

## H    VALIDATING THE REPRESENTATION POWER OF INTRA-LINKS

Although the focus of our draft is justifying the power of width in the context of intra-layer links rather than designing new architectures, our analysis theoretically suggests that an intra-linked network is a promising network structure. Therefore, it is highly necessary to validate if an intra-linked network can deliver good performance as predicted by our theory.

Before reporting experimental results, let us analyze the characteristics of an intra-lined network. First, it is straightforward to see that using intra-layer links increases a few parameters. But even if only every two neurons are intra-linked in a layer, the improvement is exponentially dependent on depth, *i.e.*, approximately $\mathcal{O}(\frac{3}{2})^k$, which is considerable when a network is deep. Therefore, they can serve as an economical yet powerful add-on to the model. Second, the complexity of computing a layer with $W$ neurons in a classical ReLU DNN is $W^2$ multiplications and $W^2$ additions while computing an intra-layer linked ReLU DNN of the same size and with every $n_i$ neurons intra-linked needs $W^2$ multiplications and $W^2 + (n_i - 1) \cdot [W/n_i] \approx W^2 + W$ additions, where $[\cdot]$ is a ceiling function, which is still quadratic. Thus, the computational cost incurred by adding intra-links is minor. When applying intra-layer links in CNNs, the links can be added between different channels. The computational cost is also minor. Third, the usage of intra-layer links may hurt the hardware optimization to some extent. However, we can design acceleration algorithms for intra-linked networks. Specifically, the acceleration of RNNs and LSTMs has been intensively investigated. We can translate ideas therein such as sequence bucketing to solve the training issues of intra-linked networks. In brief, intra-layer links are not subjected to a high computational and parametric cost and a low training speed.

We evaluate the proposed intra-layer links in both regression and classification tasks. We first conduct regression experiments on 5 synthetic polynomials, 2 widely-used public datasets, and 3 real-world datasets. Results suggest that the intra-layer links can boost the network's representation power. Second, we demonstrate the effectiveness and efficiency of intra-layer links on classification tasks using 8 tabular datasets, 2 fault diagnosis datasets, and 2 image benchmarks (CIFAR100 and Tiny-ImageNet). The experiments are implemented in Tensorflow using a CPU Intel i7-11800H processor at 2.3Hz and a GPU NVIDIA T600.

### H.1    REGRESSION EXPERIMENTS

### H.1.1    RESULTS ON SYNTHETIC DATASETS

We synthesize five elementary polynomials whose expressions are shown in Table 1. The normally-distributed noise with 0 mean and 0.5 variance is added into the synthesized signals. A total of 1,000 points are sampled from $[-3, 3]$ with an equal distance for training. The width of each layer is 8, and depths of each network are respectively set to $[3, 5, 11, 11, 11]$. The epoch is 200, the batch size is 128, and 'Adam' (Kingma & Ba, 2014) is the optimizer. The schedule for the learning rate is 'ReduceLROnPlateau'. To test the generalization accuracy of the model, 1,000 points are sampled from $[-4, 4]$, none of which appears in the training. Comparisons between fully-connected networks and intra-layer links are shown in Table 2 and Figure 10.

In Table 2, we quantitatively compare the approximation errors between intra-linked and fully-connected networks using the mean squared error (MSE). We verify two kinds of fully-connected networks: one ($s_1$) has the same width and depth as intra-linked networks; the other ($s_2$) is deeper.

Table 1: The expressions of synthetic polynomials.

| Synthetic functions | Expression |
|---|---|
| $p_1(x)$ | $x^2 + x$ |
| $p_2(x)$ | $x^3 + x^2 + x$ |
| $p_3(x)$ | $x^4 + x^3 + x^2 + x$ |
| $p_4(x)$ | $x^5 + x^4 + x^3 + x^2 + x$ |
| $p_5(x)$ | $x^6 + x^5 + x^4 + x^3 + x^2 + x$ |

Table 2: MSE values of intra-linked and fully-connected networks on synthetic experiments. #PRM denotes the number of parameters.

| Synthetic functions | Indicators | Intra-layer | Fully-connected ($s_1$) | Fully-connected ($s_2$) |
|---|---|---|---|---|
| $p_1(x)$ | MSE | 0.6616 | 1.5755 | 1.0700 |
| | #PRM | 99 | 97 | 169 |
| $p_2(x)$ | MSE | 43.6248 | 46.9811 | 45.4330 |
| | #PRM | 245 | 241 | 313 |
| $p_3(x)$ | MSE | 432.0960 | 491.0851 | 439.0270 |
| | #PRM | 683 | 673 | 745 |
| $p_4(x)$ | MSE | 17,204.2516 | 23,397.8515 | 17,936.0898 |
| | #PRM | 829 | 817 | 889 |
| $p_5(x)$ | MSE | 304,183.1563 | 334,655.9375 | 307,388.1875 |
| | #PRM | 829 | 817 | 889 |

(a)     (b)     (c)     (d)     (e)

Figure 10: Polynomial function. (a)∼(e) are separately the polynomial functions from the second order to the sixth order.

Table 3: Statistics of 5 regression tabular datasets.

| Datasets | Instances | Features | Features Type |
|---|---|---|---|
| *Boston housing* | 506 | 13 | discrete |
| *California housing* | 20640 | 8 | discrete |
| *Walmart* | 97056 | 20 | discrete |
| *Energy consumption* | 35024 | 6 | continuous |
| *Wind power* | 49166 | 5 | continuous |

Thus, $s_1$ has comparable parameters as intra-linked networks, and $s_2$ has more. We can draw two highlights from Table 2. First, when the parameters are comparable, intra-linked networks can lead to a huge improvement in MSE compared with fully-connected networks ($s_1$). For example, in approximating $p_1(x)$, the MSE decreases by 58%. Second, when the fully-connected networks go deeper ($s_2$), the intra-linked network can still perform better.

Figure 10 shows qualitatively the generalization between intra-linked and fully-connected ($s_1$) networks. We find that in $[-3, 3]$ both intra-linked and fully-connected networks agree with the original functions well. However, in approximating peripheral parts of the function like $[-4, -3]$ and $[3, 4]$, intra-linked networks outperform fully-connected ($s_1$) networks.

### H.1.2 RESULTS ON REAL-WORLD DATASETS

Furthermore, encouraged by positive results on synthetic experiments, we continue the regression experiments on 5 widely-used real-world datasets: *Boston housing*[1], *California housing*[2], *Walmart*[3],

---

[1]https://archive.ics.uci.edu/ml/machine-learning-databases/housing/housing.data

[2]https://archive.ics.uci.edu/ml/machine-learning-databases/housing/housing.data

[3]https://www.kaggle.com/c/walmart-recruiting-store-sales-forecasting

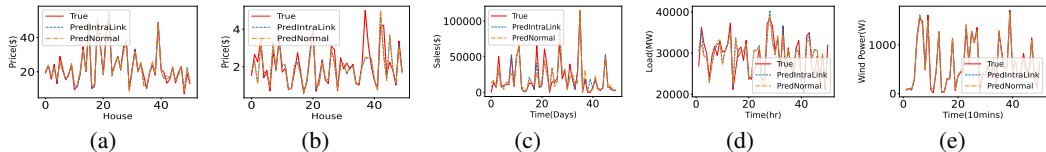

Figure 11: Real-world regression experiments. (a) Boston house price; (b) California house price; (c) Walmart sales forecasting; (d) Load consumption forecasting; (e) Wind power forecasting.

Table 4: MSE values of intra-linked and fully-connected networks on real-world datasets. #PRM denotes the number of parameters.

| Datasets | Indicators | Intra-layer | Fully-connected ($s_1$) | Fully-connected ($s_2$) |
|---|---|---|---|---|
| *Boston house price* | MSE | 5.3113 | 10.5220 | 5.9844 |
| | #PRM | 69,635 | 69,633 | 135,425 |
| *California house price* | MSE | 0.2518 | 0.2664 | 0.2631 |
| | #PRM | 69,635 | 69,633 | 135,425 |
| *Walmart sales forecasting* | MSE | 88,556,472.0 | 111,667,784 | 110,398,056 |
| | #PRM | 466,185 | 466,177 | 531,969 |
| *load consumption forecasting* | MSE | 7,922,336.5 | 8,191,562.55 | 8,191,562.55 |
| | #PRM | 923,152 | 923,137 | 988,929 |
| *Wind power forecasting* | MSE | 3,812.82 | 3,991.2966 | 3,838.5940 |
| | #PRM | 593,931 | 593,921 | 659,713 |

*Energy consumption*[4], *Wind power*[5]. The statistics of these datasets are summarized in Table 3. For small datasets, they are split into training and test sets with a ratio of 0.9:0.1. For the rest large datasets, we split them with a ratio 0.8:0.1:0.1. Each layer has 256 neurons, and depths of each network are respectively $[2, 2, 8, 15, 10]$. The epoch is 500, and the batch size is 128. Other hyperparameters are the same as synthetic experiments.

Consistent with synthetic experiments, Tabel 4 summarizes the MSE and parameters of each network. $s_1$ has the same width and depth as intra-linked network, and $s_2$ has one more layer. We also find the intra-linked networks consistently outperform fully-connected networks ($s_1$ and $s_2$) on the 5 real-world datasets. When intra-linked and fully-connected networks have comparable parameters, intra-linked networks can lead by a large margin.

Figure 11 shows visually the regression results between intra-linked and fully-connected networks on 5 datasets. By adding intra-layer within each layer, the regression performance has significant improvement in real-world regression tasks especially at peaks. For example, the Walmart sales predicted by intra-linked network at Day 10, and 19 are closer to the actual sales. For the load consumption prediction, intra-linked networks generate curves that align with the true load better, especially at 3hr and 42hr.

In brief, in regression tasks, the intra-linked network takes the lead by a large margin. Such a superiority corroborates our theoretical analysis that adding intra-layer links can boost the network's representation power.

## H.2 CLASSIFICATION

### H.2.1 RESULTS ON TABULAR DATASETS

We first investigate the effectiveness of intra-layer links on classification tasks using tabular datasets. The tabular datasets contain *Gaussian quantiles*[6], *Breast cancer*[7], *kddcup99*[8], *Wine*[9], *Banknote au-*

---

[4]https://www.kaggle.com/datasets/robikscube/hourly-energy-consumption

[5]https://www.kaggle.com/datasets/theforcecoder/wind-power-forecasting

[6]https://scikit-learn.org/stable/modules/generated/sklearn.datasets.make-gaussian-quantiles.html

[7]https://archive.ics.uci.edu/ml/datasets/Breast+cancer+Wisconsin+(Diagnostic)

[8]http://kdd.ics.uci.edu/databases/kddcup99/kddcup99.html

[9]https://scikit-learn.org/stable/modules/generated/sklearn.datasets.load wine.html

Table 5: Statistics of 10 classification tabular datasets.

| Datasets | Instances | Classes | Features | Features Type |
|---|---|---|---|---|
| *Gaussian quantiles* | 10000 | 2 | 2 | discrete |
| *Breast cancer* | 569 | 2 | 30 | discrete |
| *kddcup99* | 494021 | 23 | 41 | discrete |
| *Concentric circles* | 16000 | 4 | 2 | discrete |
| *Banknote authentication* | 1347 | 2 | 4 | discrete |
| *Heart Failure* | 299 | 2 | 12 | discrete |
| *Ionosphere* | 350 | 2 | 34 | discrete |
| *Mobile Price* | 2000 | 4 | 20 | discrete |
| *CWRU* | 2400 | 10 | 1200 | continuous |
| *Motor fault* | 3392 | 6 | 120 | continuous |

*thentication*[10], *Heart Failure*[11], *Ionosphere*[12], *Mobile Price*[13], *CWRU*[14], *Motor fault*[15]. They are publicly available from the Python scikit-learn package, UCI machine learning repository, Kaggle and so on. The *Concentric circles* and *Gaussian quantiles* are two synthetic datasets, and the rest are all real-world datasets including medical dataset, network intrusion detection dataset (*Kddcup99*), climate dataset (*Ionosphere*), mobile price dataset and fault diagnosis dataset *etc.*. All the statistics of tabular datasets are summarized in Table 5.

Similar to regression experiments, Table 6 indicates as well that intra-layer links have stronger expressivity than fully-connected networks. When evaluated on simple datasets such as *concentric circles*, *Gaussian quantiles*, and *Kddcup99*, the improvement is moderate. This is because both intra-layer links and fully-connected networks has the ability to extract features for accurate classification. In contrast, the intra-layer links perform much better on complex datasets. For example, for the fault diagnosis in *CWRU*, the gains are respectively 2.04% and 1.2%. These improvements in test accuracy substantiate the effectiveness of intra-layer links.

### H.2.2 IMAGE CLASSIFICATION EXPERIMENTS

To further illustrate the effectiveness of intra-linked networks, we conduct two image classification experiments utilizing CIFAR100 (Krizhevsky et al., 2009) and Tiny-ImageNet (Le & Yang, 2015). In the intra-linked layer, channels are equally divided into two parts, and each part is used to capture the image features. The outputs of one part are added to the other in the form of shortcuts by trainable parameters, and then concatenated. Mathematically, the input of an intra-linked layer is $[a, b]$, while the output is $[ReLU(a), ReLU(ReLU(a) + b)]$. The intra-linked network contains 1 convolutional layer, several intra-link layers, and 1 fully-connected layer with 'softmax'. We use the same hyperparameters as ResNet18 (DeVries & Taylor, 2017).

Taking ResNet18 as a benchmark, Table 7 shows that when we use fewer convolutional kernels in an intra-linked network, it has similar performance on CIFAR100 but significantly better performance on Tiny-ImageNet compared to ResNet18, which means adding intra-layer links can boost the network's representation power.

---

[10]http://archive.ics.uci.edu/ml/datasets/banknote+authentication

[11]https://archive.ics.uci.edu/ml/datasets/heart+disease

[12]http://archive.ics.uci.edu/dataset/52/ionosphere

[13]https://www.kaggle.com/datasets/iabhishekofficial/mobile-price-classification

[14]https://engineering.case.edu/bearingdatacenter/apparatus-and-procedures

[15]https://gitlab.com/power-systems-technion/motor-faults

Table 6: Test accuracy of intra-linked and fully-connected networks on tabular classification datasets. #PRM denotes the number of parameters.

| Datasets | Indicators | Intra-layer | Fully-connected ($s_1$) | Fully-connected ($s_2$) |
|---|---|---|---|---|
| *Concentric circles* | ACC | 99.93% | 99.75% | 99.43% |
| | #PRM | 61 | 60 | 132 |
| *Gaussian quantiles* | ACC | 97.2% | 96.6% | 96.1% |
| | #PRM | 554 | 546 | 618 |
| *Breast cancer* | ACC | 98.246% | 96.49% | 96.86% |
| | #PRM | 2,380,812 | 2,380,802 | 2,643,458 |
| *Kddcup99* | ACC | 99.95% | 99.94% | 99.945% |
| | #PRM | 295,961 | 295,959 | 558,615 |
| *Banknote authentication* | ACC | 100% | 98.52% | 99.26% |
| | #PRM | 3,587 | 3,586 | 266,242 |
| *Heart Failure* | ACC | 83.33% | 80.00% | 76.67% |
| | #PRM | 795,654 | 795,650 | 1,058,306 |
| *Ionosphere* | ACC | 100% | 97.14% | 94.28% |
| | #PRM | 806,918 | 806,914 | 1,069,570 |
| *Mobile Price* | ACC | 92.5% | 91.5% | 92% |
| | #PRM | 806,918 | 806,914 | 1,069,570 |
| *CWRU* | ACC | 89.12% | 87.08% | 87.92% |
| | #PRM | 2,105,873 | 2,105,866 | 2,368,522 |
| *Motor fault* | ACC | 98.53% | 97.35% | 97.64% |
| | #PRM | 2,166,287 | 2,166,278 | 2,428,934 |

Table 7: Test accuracy of intra-linked and fully-connected networks on two image datasets. #PRM denotes the number of parameters.

| Datasets | Indicators | Intra-layer | ResNet18 |
|---|---|---|---|
| CIFAR100 | ACC | 75.34% | 75.61% |
| | #PRM | 6.55M | 11.2M |
| Tiny-ImageNet | ACC | 46.60% | 42.51% |
| | #PRM | 5.07M | 11.2M |

