# OpenReview forum: "Rethink Depth Separation with Intra-layer Links"
_ICLR.cc/2024/Conference — Submitted to ICLR 2024_

### Official Review · Reviewer_vJDK · 2023-10-12

**Soundness:** 2 fair
**Presentation:** 2 fair
**Contribution:** 1 poor
**Rating:** 1
**Confidence:** 5

**Summary:**

This work studies the depth-separation in the expressive power of neural networks. By allowing intra-layer links, it's shown that such neural networks can represent particular "sawtooth" functions efficiently with a small depth.

**Strengths:**

The results are correct.

**Weaknesses:**

The results are meaningless: a very strong assumption is used to prove a very weak argument.

**Intra-link is unjustified:** the motivation of considering neural networks with intra-layer links needs justification. There should be discussion on practical architectures with intra-layer links, which is crucial for establishing the usefulness of the proposed theory. ResNet and DenseNet are not appropriate for justification: they use across-layer links which are restricted (ResNet uses only identity mapping), while the links considered in this work are intra-layer with little restriction.

**Intra-link is strong:** the network structure with intra-layer links seems too strong. In particular, we can show a much stronger argument that a two-layer intra-layer network can represent any classic neural network. In particular, suppose the target network we would like to represent has depth $k$ with width $w_1...w_k$, then it can be realized by a two-layer intra-layer network with the width of its hidden layer being $\sum_i w_i$: we partition the neurons into $k$ groups with number $w_i$, then it simulates the target network by using intra-layer links from the $ith$ group to the next one. Effectively, a single layer of an intra-link network can be a classic network.

**Weak results:** it's only proven that such strong network structure can represent particular hard-case functions used in previous depth-separation works. The minimum expectation for the conclusion is like "this new network structure can represent any deep classic networks", which I suspect can be easily obtained. However, the result only concerns the particular "saw-tooth" functions and is thereby very weak.

**Questions:**

NA

---

> ### Author Response · Authors · 2023-11-17
> **Response to vJDK**
>
> Dear Reviewer vJDK:
>
> **[Q1] Intra-link is unjustified.**
>
> The depth separation theory suggests that depth is significantly more powerful than width, as reducing the depth will lead to the exponential width in expressing some functions. However, their comparison is always based on the standard fully-connected neural networks. In reality, most deep networks use shortcuts such as ResNet and DenseNet. There is a gap between the depth separation theory and the networks that are often deployed. Our motivation is to fill this gap by comparing the power of depth and width in the context of shortcuts, which is a more realistic setting. Particularly, we would like to investigate the question: Is the width still that weak in the new setting? If we use extra-layer links such as ResNet, critiques may arise that in an unrolled view, ResNet is a simultaneously wide and deep network, which makes it hard to examine the width. Therefore, we choose to add intra-layer links, which does not change the width. Moreover, from a symmetric viewpoint, adding extra-layer links can save the depth, and adding intra-layer links can save the width.
> We have emphasized our motivation in this revision.
>
>
> **[Q2] Intra-link is strong: the network structure with intra-layer links seems too strong. In particular, we can show a much stronger argument that a two-layer intra-layer network can represent any classic neural network. In particular, suppose the target network we would like to represent has depth $k$ with width $w_1,\ldots,w_k$, then it can be realized by a two-layer intra-layer network with the width of its hidden layer being $\sum_i w_{i}$: we partition the neurons into $k$ groups with number $w_i$, then it simulates the target network by using intra-layer links from the $i$-th group to the next one. Effectively, a single layer of an intra-link network can be a classic network.**
>
>
> What you propose is unfortunately infeasible.
>
> A two-layer ReLU network can express any $\mathbb{R} \to \mathbb{R}$ ReLU network. But the problem is efficiency. If we use a shallow network to express a deep network, the width we need is exponential, which is the key idea of depth separation. We also know that a two-layer intra-layer network can represent any classic neural network, as long as the width is sufficiently large. However, that is irrelevant to the depth separation theory. Our work is to show this gap can be greatly reduced if we use intra-layer links.
>
> Besides, we are afraid that your assertion is incorrect. Based on the tight bound analysis in our analysis, if the target network has depth $k$ with width $w_1,\ldots,w_k$, the width we need is approximately $\prod_{i} w_{i}$ for a two-layer network, instead of  $\sum_{i} w_{i}$ as you asserted.
>
> From the perspective of the number of parameters, adding intra-layer links in a fully-connected network is no more than doubling the number of depth, which means that intra-links are not strong assumptions. But what we have realized is indeed significant. Based on [Telgarsky 2015], doubling the depth of a fully-connected network only reduces the width from $w$ to $w^{1/2}$. However, intra-linking $n$ neurons within a layer can bring an exponential saving for the width based on our dedicated construction.
>
> We sincerely hope that you can consider raising your score, as our work is not as straightforward as you think.
>
> **[Q3] Weak results: it's only proven that such strong network structure can represent particular hard-case functions used in previous depth-separation works. The minimum expectation for the conclusion is like "this new network structure can represent any deep classic networks", which I suspect can be easily obtained. However, the result only concerns the particular "saw-tooth" functions and is thereby very weak.**
>
> As is claimed in  [Q2], our goal is to extend current depth separation theory to shortcut architecture. Our result shows depth separation can be modified with intra-layer links, instead of the ability to represent any networks. The statement of depth separation is that there exists a function representable by a deep network which cannot be represented by a shallow network whose width is lower than a large threshold. So, some specific functions are needed to support depth separation theory.
>
> Furthermore, the sawtooth function is not a hardcase function. Actually, it is a fundamental function in network approximation theory (c.f., [1],[2]). Therefore, most of the relevant approximation results can be easily extended to intra-linked structures, which can greatly save width compared to conventional networks.
>
>
> Reference:
> [1]Lu, Jianfeng, et al. "Deep network approximation for smooth functions." SIAM Journal on Mathematical Analysis 53.5 (2021): 5465-5506.
> [2]Yarotsky D. Error bounds for approximations with deep ReLU networks[J]. Neural Networks, 2017, 94: 103-114.

---

> > ### Comment · Reviewer_vJDK · 2023-11-22
> >
> > Thanks for the reply.
> >
> > When we talk about width/depth/layer, these words have to be put under a specific context for a rigorous definition. For the classic network structures, when we say depth separation, we mean there exists a deep network, when approximated by a shallow one it requires exponentially more parameters.
> >
> > Now, let me define a new network structure, where I call the whole network a "layer", then I claim every network in this structure has one layer and thus magically breaks the depth separation.
> >
> > Clearly this is wrong claim. Why? Because I made the straw man fallacy: the meaning of depth is different in the two network structures.
> >
> > Intra-link network has a similar issue: a two-layer intra-link network contains classic deep networks as a subset, therefore it makes no sense to compare depth between intra-link network and classic network. The way you compare the depth is misleading.
> >
> > The theme of depth separation in classic network is polynomial diversity (function addition brought by width) versus exponential efficiency (function composition brought by width). A shallow network has to use exponentially more parameters by using function addition to imitate function composition which it's not good at. This is the reason depth separation theory only concerns the hardness of approximating certain functions, like the saw-tooth, which naturally fits depth more than width. However, the way intra-link network breaks the depth separation is by including deep classic networks as a subset, this is a tautology.

---

### Official Review · Reviewer_b7mN · 2023-10-28

**Soundness:** 3 good
**Presentation:** 4 excellent
**Contribution:** 2 fair
**Rating:** 6
**Confidence:** 2

**Summary:**

The paper introduces a class of models with the goal of increasing the representational capabilities of shallow networks. It does so by introducing "intra-layer links", i.e. each neuron is directly linked (through addition) to other neurons in the same layer. In particular, the paper studies the cases where (1) each neuron depends on its proceedings neurons, and (2) every two neurons are linked. They show that this model can approximate the sawtooth function with a significant reduction in width compared to a model of the same depth without intra-layer links. In particular, the reduction in width is exponential. The authors imply that these results should spark reflection in the field of depth separation theory, where depth is often assumed to play a crucial role in efficiently approximating certain classes of functions such as the sawtooth function. The authors also show preliminary results on common datasets (such as CIFAR100 and Tinyimagnet and others).

**Strengths:**

1. The paper is well-written, the authors nicely present the background concepts on depth separation required to understand the manuscript.
2. The differences in terms of extra-linked neural network vs intra-linked nets are nicely explained through Theorem 4 and 6, where it is shown the upper bound on the number of different pieces that each variant can represent (a factor of $w_i + 1$ per layer for extra-linked, and $2^{n_i}-1/n w_i + 1$ for intra-linked.
3. Theorem 10 nicely explains the effect of intra-linked layers in terms of function approximation. Lemma 11 specializes Theorem 10 to the sawtooth function. In general, I enjoyed the cleanness and preciseness of the exposition of the theorems.
4. Although I am not entirely familiar with the field of approximation theory, I would imagine this paper's idea of proposing novel architecture with desirable function approximation properties to spark interest in the community.

**Weaknesses:**

1. I appreciate that the authors put forward an extensive explanation of why they think that the intra-linked layer is different from stacking layers. From an information propagation perspective, however, I still struggle to agree with the authors. The quantity $\tilde{f}_i^{(j)}$ as in the equation of Notation 2 depends on the previous $p < j$ preactivations $g_i^{(0)}, \dots, g_i^{(p-1)}$. This is similar to add skip connection, where the information is residually added on a neuron-by-neuron way to form the representation of the next neuron. Of course, as it authors point out, it is not the standard way depth is thought in neural networks, and there are differences in terms of the absence of extra weights, and different handling of nonlienarities, which also has an impact on the function class representable and the mechanics of producing linear pieces.
2. Related to (1). I am not entirely convinced of why the authors have centered the story around the role of width vs depth. I think the intra-layer mechanism is something out of the scope of depth-vs-width in the way fully connected neural networks are treated in the dept separation theory. In the depth separation theory, the objective is to study the roles of width and depth in the "natural" way they are treated in the literature. In the paper, the authors design a new architecture in which each layer has something that resembles depth, and claim that the paper's objective is to "inspire further contemplation and discussion regarding the depth separation theory", which in my honest view is misleading. I would like the authors' clarification on this point.
3. The potential of the architecture for practical usage seems very limited. The computation is inherently sequential and thus cannot make use of existing hardware optimization through parallelization.

**Questions:**

See weaknesses.

---

> ### Author Response · Authors · 2023-11-17
> **Response to b7mN**
>
> Dear Reviewer b7mN:
>
> We would like to thank you for your recognition of the presentation and novelty in our work. Here, we address your concerns and answer your questions. We look forward to more discussions with you!
>
> **[Q1] I appreciate that the authors put forward an extensive explanation of why they think that the intra-linked layer is different from stacking layers...which also has an impact on the function class representable and the mechanics of producing linear pieces.**
>
> Thanks for these insightful comments. Let us use a simpler example to illustrate this point. Suppose we have a fully-connected network with width=$w$ and depth=$k$ to express a sawtooth function, we have two choices to reduce the width to express the same sawtooth function. The first is to increase the depth of this fully-connected network from $k$ to $nk$, based on [Telgarsky 2015], the width is reduced from $w$ to $w^{1/n}$. The second is to intra-link $n$ neurons in the network, which can bring an exponential saving for the width based on our construction. Therefore, our result is indeed highly non-trivial and provides a new insight into the depth separation theory.
>
>
> Defining the width and depth of a network with shortcuts is indeed tricky. For example, the famous ResNet can be unfolded into a simultaneously wide and deep network whose width and depth are the same. However, few researchers will take the ResNet as a wide network and attribute its success to width instead of depth, thereby vetoing the success of deep learning. To avoid such a conflict,
> in our draft, we define the width and depth as
>
> **[Width and depth of intra-linked networks, (Fan-Lai-Wang, JMLR2023)]** Given an intra-linked network $\mathbf{\Pi}$, we delete the intra-layer links layer by layer to make the resultant network $\mathbf{\Pi}'$ a standard fully-connected network, which means it has no isolated neurons and shortcuts. Then, we define the width and depth of $\mathbf{\Pi}$ to be the same as the width and depth of $\mathbf{\Pi}'$.
>
> Such a definition well aligns with our conventional understanding of width and depth, compared to using the broadest concatenating neurons as the width and the longest path as the depth.
>
>
> **[Q2] Related to (1). I am not entirely convinced of why the authors have centered the story around the role of width vs depth. I think the intra-layer mechanism is something out of the scope of depth-vs-width in the way fully connected neural networks are treated in the dept separation theory...**
>
> Thanks for these insightful comments. The depth separation theory suggests that depth is significantly more powerful than width, as reducing the depth will lead to the exponential width in expressing some functions. However, their comparison is always based on the standard fully-connected neural networks. In reality, most deep networks use shortcuts such as ResNet and DenseNet. There is a gap between the depth separation theory and the networks that are often deployed. Our motivation is to fill this gap by comparing the power of depth and width in the context of shortcuts, which is a more realistic setting. Our theoretical results suggest that with intra-linked networks, the width needed is greatly reduced. Thus, we derive a new relationship between width and depth in a different setting, which provides a different perspective than what is suggested in the depth separation theory.
>
> In this revision, we have clarified the gap between the existing depth separation theory and reality and explained our motivation to provide a new relationship between width and depth in a more realistic setting.
>
>
> **[Q3] The potential of the architecture for practical usage seems very limited. The computation is inherently sequential and thus cannot make use of existing hardware optimization through parallelization.**
>
> Thanks for this suggestion. We agree with you that the usage of intra-layer links may hurt the hardware optimization to some extent. However, we still think intra-linked networks are a promising architecture from two aspects:
>
> - Our experiments on 5 synthetic datasets, 15 tabular datasets, and 2 image benchmarks demonstrate that intra-linked networks can achieve better or comparable performance with fewer parameters. In memory-constraint scenarios, the intra-linked networks may be preferred.
>
> - We can design acceleration algorithms for intra-linked networks. Specifically, the acceleration of RNNs and LSTMs has been intensively investigated. We can translate ideas therein such as sequence bucketing [1] to solve the training issues of intra-linked networks.
>
> [1] Khomenko, V., Shyshkov, O., Radyvonenko, O., and Bokhan, K. (2016, August). Accelerating recurrent neural network training using sequence bucketing and multi-gpu data parallelization. In 2016 IEEE First International Conference on Data Stream Mining and Processing (DSMP) (pp. 100-103). IEEE.

---

> > ### Comment · Reviewer_b7mN · 2023-11-22
> > **Answer to Rebuttal**
> >
> > I thank the authors for clarifying my questions, especially regarding the difference between depth and intra-layer links. Indeed it can be tricky to define the concept of depth when such modifications are performed. In my view, the fact that each neuron depends on the previous neurons in the same layer is a feature typically attributed to depth. However, I also see the authors' point that in terms of function approximation intra-layer links have different properties from the classical view of depth. I keep my score for now.

---

### Official Review · Reviewer_45bo · 2023-10-31

**Soundness:** 3 good
**Presentation:** 2 fair
**Contribution:** 2 fair
**Rating:** 5
**Confidence:** 2

**Summary:**

This paper discusses theoretically about whether width of deep neural networks is always significantly weaker than depth. It introduces intra-layer links, and shows that width can also be powerful when armed with them.

**Strengths:**

- The paper recognizes an important limitation of depth-width comparison theory - it is always based on the fully-connected networks. It adds interesting and novel points to the power of width.

- The paper includes comprehensive theorems and proofs regarding its points, and also includes both synthesized and real-world experiments.

**Weaknesses:**

- The significance of theoretical contribution should be made more clear in the paper. As a reviewer outside of the learning theory field, I cannot conclude how much the improvement is compared to prior works.

- Based on my understanding, the depth of a neural network is the number of neurons on the longest path from the input to the output, and thus adding an intra-link is equivalent to inserting a new layer, which doubles the depth of the neural network. Hence, I do not agree with the claims in the paper saying that the width can also be powerful when armed with intra-links.

**Questions:**

See weaknesses.

---

> ### Author Response · Authors · 2023-11-17
> **Response to 45bo**
>
> Dear Reviewer 45bo:
>
> We would like to thank you for your recognition of the strengths in our work. Here, we address your concerns and answer your questions. We look forward to more discussions with you!
>
> **[Q1] The significance of theoretical contribution should be made more clear in the paper. As a reviewer outside of the learning theory field, I cannot conclude how much the improvement is compared to prior works.**
>
> Thanks for your advice! The improvement compared to prior works is indeed significant, which can be summarized as follows:
>
> -  With inner links, a network of depth $k$ and width $w$ has $\mathcal{O}(\frac{2^{wk}}{w^k})$ times amount of pieces compared to conventional networks.
>
> - When expressing a sawtooth function represented by a fully-connected $2k^2+1$-layer ReLU DNN with $w$ nodes in each layer, a classical network with $(k + 1)$ needs width at least $w^k$ neurons in each layer. However, with inner links, a network with $(k + 1)$ only needs no more than $k\cdot log_{2} w+2$ neurons in each layer to express such functions. The width is exponentially reduced.
>
> **[Q2] Based on my understanding, the depth of a neural network is the number of neurons on the longest path from the input to the output, and thus adding an intra-link is equivalent to inserting a new layer, which doubles the depth of the neural network. Hence, I do not agree with the claims in the paper saying that the width can also be powerful when armed with intra-links.**
>
> We agree with you that a network with intra-layer links can be unfolded into a dense network with a special arrangement. Actually, any network with shortcuts can be unfolded, too. However, inserting intra-layer links is intrinsically different from increasing the depth. Following your thoughts, can doubling the depth of a fully-connected network make the width exponentially reduced? The answer is unfortunately negative. From the perspective of the number of parameters, adding intra-layer links in a fully-connected network is no more than doubling the number of depth. Based on [Telgarsky 2015], doubling the depth of a fully-connected network only reduces the width from $w$ to $w^{1/2}$. However, intra-linking $w$ neurons within a layer can bring an exponential saving for the width based on our construction. Therefore, our result is indeed highly non-trivial and provides a new insight into the depth separation theory.
>
> In fact, allowing increasing depth, the deeper feedforward network has a larger function class than a shallow intra-linked network, and the function class of our intra-linked network is a proper subset of a much deeper fully-connected network. However, given the same width and depth, our intra-linked network has more expressive power (i.e., number of pieces, VC dimension, than a feedforward network per neuron or per parameter.
>
> For example, we consider the function class represented by a 2-layer ReLU DNN with width 2. The function class of such networks without links has at most 3 pieces, and its VC dimension is 3. However, the function class of inner linked networks has at most 4 pieces, and its VC dimension is 4. This phenomenon can be seen as an analog to the comparison between CNNs and fully-connected NNs. The function classes of CNNs are just subsets of the function classes of fully-connected NNs with some further restrictions on the weights. However, CNNs usually have more expressive power per parameter and achieve better results in practice.
>
> In addition, defining the width and depth of a network with shortcuts is indeed tricky. For example, the famous ResNet can be unfolded into a simultaneously wide and deep network whose width and depth are the same. However, few researchers will take the ResNet as a wide network and attribute its success to width instead of depth, thereby vetoing the success of deep learning.
> In our draft, we define the width and depth as
>
> **[Width and depth of intra-linked networks, (Fan-Lai-Wang, JMLR2023)]** Given an intra-linked network $\mathbf{\Pi}$, we delete the intra-layer links layer by layer to make the resultant network $\mathbf{\Pi}'$ a standard fully-connected network, which means it has no isolated neurons and shortcuts. Then, we define the width and depth of $\mathbf{\Pi}$ to be the same as the width and depth of $\mathbf{\Pi}'$.
>
> Such a definition well aligns with our conventional understanding of width and depth, compared to using the broadest concatenating neurons as the width and the longest path as the depth.
>
> Lastly, we agree with you that we might overclaim the power of width since the saving of width is due to the intra-layer links. In this revision, we have lowered our tone from contending the width is also powerful to contending that the width is not that weak. Earlier depth separation theory suggests that an exponential width is comparable to the constant depth, while our result suggests that in the context of intra-layer links, the exponential width is not necessary.

---

### Official Review · Reviewer_Nadz · 2023-11-03

**Soundness:** 3 good
**Presentation:** 3 good
**Contribution:** 3 good
**Rating:** 5
**Confidence:** 3

**Summary:**

The paper studies depth-width tradeoffs in neural networks where intralinks within the layers are allowed. The main claim to fame is that intralinks give rise to unexpected behaviour as far as experessivity depth-width bounds are concerned. In particular, by allowing intralinks between neurons of the same layer, the authors show that the network can efficiently represent certain highly oscillatory functions that have been previously used to show lower bounds for the width of shallow neural networks.

**Strengths:**

+I believe the paper studies an interesting question regarding the expressivity of neural nets with added intra links.

+The conceptual contribution that intra links significantly help the representation capabilities of neural nets w.r.t. their width requirements is nice.

**Weaknesses:**

- I found most of the results and derived bounds relatively straightforward given the works of Telgarsky, Montufar etc.

- I have trouble understanding why intra links is not effectively the same as adding extra layers to the network. If this is the case, I believe the results are not surprising as they should follow with simple variations from known depth-width bounds.

-the architecture with intralinks as shown in Fig. 1c, I am not sure it has been popular or widely used in empirical studies. So people that are more interested in experimental performance, I am not sure how they will interpret these results for networks they don't really use.

**Questions:**

-Can the authors provide simple examples for why an intralinked network cannot directly be used to "simulate" a deep network? Can they elaborate more on their explanation on page 5 top?

-What is the main novelty in the constructions? Let's for now agree that intralinked networks are interesting and there is some conceptual contribution there. The effect they have in representing sawtooth functions should be relatively straightforward so I can't understand exactly the technical novelty of the paper.

---

> ### Author Response · Authors · 2023-11-17
> **Response to Nadz (Part I)**
>
> Dear Reviewer Nadz:
>
> We would like to thank you for your recognition of the conceptual contribution to our work. Here, we address your concerns and answer your questions.
>
> **[Q1] I found most of the results and derived bounds relatively straightforward given the works of Telgarsky, Montufar etc.**
>
> Our results are non-trivial extensions of previous works. The method we estimate the bound of networks is completely different from previous works of Aurora, Montufar etc. On the one hand, we identify conditions for the tightness of the bound, which has been proven to be stronger than existing results. Specifically, in the activation step, we distinguish the existing and newly generated breakpoints to avoid repeated counting, and then in the following pre-activation step, we maximize the oscillation to yield the most pieces after the next activation. On the other hand, the construction of functions in our work, i.e., constructing oscillations by preserving existing breakpoints and splitting each piece into several ones, is generic in analyzing networks. Because of the decoded mechanisms of generating more pieces and oscillations,
>
> - Our bound estimate for conventional networks is much tighter than previous results;
>
> - We also give bound analysis for linked networks, which is applicable for both inner and outer links;
>
> - We also show the tightness of our bound with explicit construction.
>
> - We can construct an intra-linked network whose width needed can be exponentially reduced.
>
> **[Q2] I have trouble understanding why intra links is not effectively the same as adding extra layers....**
>
> Sorry for the confusion. Any network with shortcuts can be reshaped into a standard fully-connected network. For example, the famous ResNet can be unfolded into a simultaneously wide and deep network whose width and depth are the same. However, few researchers will take the ResNet as a wide network and attribute its success to width instead of depth, thereby vetoing the success of deep learning. The use of intra-layer links is fundamentally different from adding depth. From the perspective of the number of parameters, adding intra-layer links in a fully-connected network is no more than doubling the number of depth. Based on the depth separation theory [Telgarsky 2015], when doubling the depth of a fully-connected network, the width is only reduced from $w$ to $w^{1/2}$. However, our saving for the width is exponential instead of polynomial, which means that adding intra-layer links has an essentially different mechanism from adding depth. We have illustrated this point in this revision.
>
>
> We think a much deeper feedforward network has a larger function class than a shallow intra-linked network, and the function class of our intra-linked network is a proper subset of a much deeper feedforward network. However, given the same width and depth, our intra-linked network has more expressive power (i.e., number of pieces, VC dimension, than a feedforward network per neuron or per parameter. Since the function class is different, the result is of course not straightforward.
> For example, we consider the function class represented by a 2-layer ReLU DNN with width 2. The function class of such networks without links has at most 3 pieces, and its VC dimension is 3. However, the function class of inner linked networks has at most 4 pieces, and its VC dimension is 4.
>
> This phenomenon can be seen as an analog to the comparison between CNNs and fully-connected NNs. The function classes of CNNs are just subsets of the function classes of fully-connected NNs with some further restrictions on the weights. However, CNNs usually have more expressive power per parameter and achieve better results in practice.
>
> **[Q3] The architecture with intra-links as shown in Fig. 1c, ...**
>
> Exploring new and powerful network architectures such as the invention of ResNet has been the mainstream research direction in deep learning in the past decade. Although shortcuts have been widely adopted in network design, to the best of our knowledge, we are the first to consider adding shortcuts within a layer in a fully-connected network. Like the residual connections, we spotlight that no matter how many neurons are linked, the improvement of representation power by intra-layer links increases no trainable parameters for a network. Thus, the intra-layer link is an extremely economical add-on to the model, which has the great potential of enhancing model compactness. Furthermore, our analysis demonstrates that if intra-linking more neurons in a layer, the improvement in the network expressivity can be exponential. Finally, We also empirically confirm the good regression and classification performance of networks with intra-layer links via 5 synthetic datasets, 15 tabular datasets, and 2 image benchmarks in Appendix H). Therefore, we think our comprehensive theoretical analysis and encouraging experimental results can engage people focusing on practical applications.

---

> > ### Author Response · Authors · 2023-11-17
> > **Response to Nadz (Part II)**
> >
> > **[Q4] What is the main novelty in the constructions? ...**
> >
> > The technical novelty of our draft is threefold:
> >
> > - The construction itself we provide here is meaningful. Simply estimating the bound is not sufficient to justify the depth separation, since it is unknown if the deep network can also represent some function. Earlier results like [3] lack the construction analysis.
> >
> > - As we mentioned earlier, our construction realizes the exponential saving, while according to the previous construction [Telgarsky 2015], simply adding more layers in the fully connected network only leads to the polynomial saving.
> >
> > - Efficiently representing the sawtooth function is important. In deep learning approximation theory, many theoretical derivation such as [1] relies on [2] whose core is to express a power function by a sawtooth function. This sawtooth function is realized by a fully-connected network. Since using intra-layer links can express the sawtooth function much more efficiently, many theoretical derivations can be further improved.
> >
> >
> > [1] Lu, Jianfeng, et al. Deep network approximation for smooth functions.
> >
> > [2] Yarotsky D. Error bounds for approximations with deep ReLU networks.
> >
> > [3] Arora, R., Basu, A., Mianjy, P., Mukherjee, A. Understanding deep neural networks with rectified linear units. ICLR.

---

### Official Review · Reviewer_4fqv · 2023-11-08

**Soundness:** 3 good
**Presentation:** 2 fair
**Contribution:** 3 good
**Rating:** 6
**Confidence:** 4

**Summary:**

Authors consider a relatively new architectural tool for neural networks through what is known as intra-layer links, where one can have a link inside a hidden layer between two neurons and then re-visit the depth separation problem in this context. They consider the depth separation between a $k^2$ vs $k$ hidden layer networks as considered by Telgarsky (hard instances/functions) and show that a shallow network with every two neurons in each hidden layer linked via intra-layer links requires at most $\log(w)k+2$ width, while a shallow network of a standard feedforward network would require at least $w^k$ width, hence showing a possibility of gaining additional expressive power with shallow networks. They also perform experiments on synthetic and real data which seem to agree with their theoretical findings.

**Strengths:**

1) Provides a way to potentially reduce the number of parameters, i.e, use shallower networks with intra-links, since these links do not contain any trainable parameters.

2) They also show that ResNets and DenseNets have (potentially) higher representation power by looking at the maximum number of pieces that can be generated by them and compare to a standard DNN.

**Weaknesses:**

1) These intra-links are not used at all in practice and could end up being quite artificial, although ResNets (which have extra-layer links) are proven to be useful in practice.

2) I feel that the authors could do a better job in explaining possible trade-offs of adding these links (see questions for more details).

**Questions:**

1) The meaning of zero-points must be defined clearly.

2) Are Bi-directional links of any help?

3) What are the trade-offs for adding as many links as possible for a fixed architecture? Meaning why is one not incentivized to add links between all possible neurons?

4) In the experiments could you indicate how many links were added and is there an effect on optimization/generalization/efficiency on increasing the number of links?

5) When is it better to add extra-layer link vs intra-layer link?

6) Are there are more experiments where authors compare the performance to ResNets?

---

> ### Author Response · Authors · 2023-11-18
> **Response to 4fqv (Part I)**
>
> Dear Reviewer 4fqv:
>
> We would like to thank you for your recognition of the strengths in our work. Here, we address your concerns and answer your questions. We look forward to more discussions with you!
>
> **[Q1] These intra-links are not used at all in practice and could end up being quite artificial, although ResNets (which have extra-layer links) are proven to be useful in practice.**
>
> The contributions of this draft are mainly theoretical. The depth separation theory suggests that depth is significantly more powerful than width, as reducing the depth will lead to the exponential width in expressing some functions. However, their comparison is always based on the standard fully-connected neural networks. In reality, most deep networks use shortcuts such as ResNet and DenseNet. There is a gap between the depth separation theory and the networks that are often deployed. Our motivation is to fill this gap by comparing the power of depth and width in the context of shortcuts, which is a more realistic setting. Our theoretical results suggest that with intra-linked networks, the width needed is greatly reduced. Thus, we derive a new relationship between width and depth in a different setting, which provides a different perspective than what is suggested in the depth separation theory.
>
> At the same time, as a side product of our theoretical analysis, although the intra-linked network is not popular now, it has the potential to be a well-performing architecture. First, it has good expressive power. Second, the intra-layer link is an extremely economical add-on to the model, which has the great potential of enhancing model compactness. Finally, We also empirically confirm the good regression and classification performance of networks with intra-layer links via 5 synthetic datasets, 15 tabular datasets, and 2 image benchmarks in Appendix H). Therefore, we believe people focusing on practical applications should also be interested in this novel structure. We will intensively investigate this architecture in the future.
>
> **[Q2] The meaning of zero-points must be defined clearly.**
>
> Sorry for the confusion. We should define that the zero-point $x\in\mathbb{R}$ of a function $f$: $\mathbb{R}\to\mathbb{R}$ satisfies $f(x)=0$. We have added this definition in this revision.
>
> **[Q3] Are Bi-directional links of any help?**
>
> Thanks for your helpful suggestion. Bidirectional links are an interesting direction to explore. In the seminal Hopfield network, bidirectional links are used to construct the memory and query the memory. We notice that the decoded mechanism of producing pieces and oscillations cannot be directly extended into analyzing bidirectional links. We will consider this idea in our future work.
>
> **[Q4] What are the trade-offs for adding as many links as possible for a fixed architecture? Meaning why is one not incentivized to add links between all possible neurons?**
>
> We notice that there are two kinds of trade-offs arising from adding intra-layer links:
>
> - The computational complexity: let us analyze the characteristics of an intra-lined network. First, it is straightforward to see that using intra-layer links increases a few parameters. But even if only every two neurons are intra-linked in a layer, the improvement is exponentially dependent on depth, \textit{i.e.}, approximately $\mathcal{O}(\frac{3}{2})^{k}$, which is considerable when a network is deep. Therefore, they can serve as an economical yet powerful add-on to the model. Second, the complexity of computing a layer with $W$ neurons in a classical ReLU DNN is $W^2$ multiplications and $W^2$ additions while computing an intra-layer linked ReLU DNN of the same size and with every $n_i$ neurons intra-linked needs $W^2$ multiplications and $W^2+(n_i-1)\cdot[W/n_i]\approx W^2+W$ additions, where $[\cdot]$ is a ceiling function, which is still quadratic. Thus, the computational cost incurred by adding intra-links is minor. When applying intra-layer links in CNNs, the links can be added between different channels. The computational cost is also minor. In brief, intra-layer links are not subjected to a high computational and parametric cost.
>
> - Hardware optimization through parallelization: The usage of intra-layer links may hurt the hardware optimization to some extent. However, we can design acceleration algorithms for intra-linked networks. Specifically, the acceleration of RNNs and LSTMs has been intensively investigated. We can translate ideas therein such as sequence bucketing to solve the training issues of intra-linked networks.
>
> We have illustrated these trade-offs in this revision.

---

> > ### Author Response · Authors · 2023-11-18
> > **Response to 4fqv (Part II)**
> >
> > **[Q5] In the experiments could you indicate how many links were added and is there an effect on optimization/generalization/efficiency on increasing the number of links?**
> >
> > Thanks for this interesting question. The number of links changes with different tasks, and there are two kinds of links in each model. Normally, there are 3$\sim$ 10 layers. The first connection is between two layers same as MLP. The second connection is between neurons in the same layer. The neurons in the same layer are evenly divided into two parts and connected like shortcuts.
> >
> > Because the neurons in the same layer are evenly divided into two parts, if there are enough neurons in one layer, the generalization and efficiency will increase. But if the neurons in one layer are limited, the number of intra-links needs to be limited because the feature extraction needs enough neurons.
> >
> >
> > **[Q6] When is it better to add extra-layer link vs intra-layer link?**
> >
> > Thanks for this interesting question. In practice, we will suggest to simultaneously use extra-layer and intra-layer links. The intra-layer link is an efficient add-on, which is also compatible with extra-layer links such as ResNet. Our experiments on two image benchmarks also show that adding intra-layer links is beneficial to ResNets.
> >
> > **[Q7] Are there more experiments where authors compare the performance to ResNets?**
> >
> > Thanks for this question. As we mentioned earlier, the main contribution of this draft is theoretical. In the future, we will systematically evaluate the empirical performance of the intra-linked networks, including comparing the performance to ResNets and other shortcut networks.

---

> ### Comment · Reviewer_4fqv · 2023-12-04
>
> Dear authors,
>
>   Thanks for the responses, but I would keep my current score, as I still feel the best way to show the strength of the approach is maybe to use experiments that compares accuracy, # of parameters and training time against standard feed-forward networks in such a way that say the # of trainable parameters remains a constant. This might give more evidence in support of shallower networks as opposed to do deep networks and it would be interesting to see the performance (both training time and test accuracy). As the other reviewers point out the theory is limited as it focuses on improving some previously known "hard instances" for standard deep networks, which points to evidence that the idea proposed by the authors may be useful. Thus showing sufficient experimental evidence on standard benchmarks and clearly explaining the trade-offs involved could help.

---

### Meta-Review · Area_Chair_VSnM · 2023-12-07

**Metareview:**

The paper studies depth-width tradeoffs in neural networks where intralinks within the layers are allowed. By allowing intralinks between neurons of the same layer, the authors show that the network can efficiently represent certain highly oscillatory functions that have been previously used to show lower bounds for the width of shallow neural networks in a paper by Telgarsky. It seems that the paper has merits and the idea of intralink is quite interesting and might be beneficial as pointed by 4fqv. However, it also seems that the theory of the paper is quite limited and more experiments should be added to strengthen the submission, so in its current form it is below the acceptance bar.

**Justification For Why Not Higher Score:**

Limited math. The paper might benefit from further experiments.

**Justification For Why Not Lower Score:**

NA

---

### Decision · Program_Chairs · 2024-01-16

Reject